# Ab initio mechanism revealing for tricalcium silicate dissolution

Yunjian Li [1], Hui Pan [1,2], Qing Liu[1], Xing Ming[1] & Zongjin Li [1 ✉]

Dissolution of minerals in water is ubiquitous in nature and industry, especially for the calcium silicate species. However, the behavior of such a complex chemical reaction is still unclear at atomic level. Here, we show that the ab initio molecular dynamics and metadynamics simulations enable quantitative analyses of reaction pathways, thermodynamics and kinetics of the calcium ion dissolution from the tricalcium silicate ($Ca_3SiO_5$) surface. The calcium sites with different coordination environments lead to different reaction pathways and free energy barriers. The low free energy barriers result in that the detachment of the calcium ion is a ligand exchange and auto-catalytic process. Moreover, the water adsorption, proton exchange and diffusion of water into the surface layer accelerate the leaching of the calcium ion from the surface step by step. The discovery in this work thus would be a landmark for revealing the mechanism of tricalcium silicate hydration.

[1] Institute of Applied Physics and Materials Engineering, University of Macau, Macao SAR 999078, P. R. China. [2] Department of Physics and Chemistry, Faculty of Science and Technology, University of Macau, Macao SAR 999078, P. R. China. ✉email: zongjinli@um.edu.mo

Exploring the kinetics of dissolution and dynamic properties at the water/solid interface on the atomic scale is of great significance to understand the natural process and instruct the industrial production at macroscopic scale. This has been at the heart of numerous research fields, such as geochemistry[1,2], drug release[3], water treatment[4], and degradation of catalysis[5]. Calcium silicate is an essential constituent in many natural minerals and has been used in a variety of fields from building materials[6–9] to pharmaceutical products[3,10]. Because of its bioactivity, biocompatibility and hydraulic nature, it is also a candidate for drug delivery[11,12], filling and regeneration materials in dentistry[13,14] and bone tissue[15]. Above all, its application in cement is of great interest due to huge amount of usage in world widely. Tricalcium silicate ($Ca_3SiO_5$) is the main and most reactive calcium silicate species in ordinary Portland cement (OPC)[6]. It is well known that the cement hydration is stimulated by the dissolution of calcium ions from the $Ca_3SiO_5$ surfaces accompanied by the precipitation of lamellar calcium-silicate-hydrate (C–S–H), which is responsible for the cohesivity, durability and mechanical properties of concrete[16].

The $Ca_3SiO_5$ hydration exhibits clear stages (initial, induction, acceleration, and deceleration) and is governed by multiple coupled parameters diverging in different time scales (from fs to years) and space scales (from nanoscale to macroscale), which is extremely complex to depict precisely. The experimental studies found that during the dissolution process the surface topography undergoes a complicated transformation with the formation of etch pits, point defects and screw dislocation[17]. Besides, the hydrated silicate species above the surfaces reconstruct with the remaining Ca ions after the detachment of Ca ions[18,19]. In general, the dissolution rate is well accepted to be affected by the grain particles size, overall reactive surface area, temperature, components of solution and dislocations on the solid surface[20] on the macroscopic scale. Alongside these, the global dissolution rate is also controlled by the slowest step, which depends on the individual stage during reaction. However, the case would be more intricate for $Ca_3SiO_5$ due to the coupling effect with the precipitation of hydrate product. It has been observed the dissolution rate of $Ca_3SiO_5$ is extremely fast initially and then decreases dramatically from the highest to the lowest[21]. The reasons for this phenomenon are still on debate. Firstly, the hydroxylation prior to dissolution may stabilize the surface and therefore lower the solubility of $Ca_3SiO_5$, as is the case for other minerals[20]. Furthermore, the dissolution theory[17] implies the driving force for the initial swift dissolution rate is the high degree of undersaturation as it is energetically favorable for etch pits to form. When the composition of the solution is very close to the solubility equilibrium of $Ca_3SiO_5$, the etch pits no longer form and even step retreat, thus limiting the dissolution rate rather severely[22], like the natural weathering and other mineral hydration[23]. Moreover, there may also be an electrical double layer[24] and a metastable hydrate phase barrier[21] formed on the $Ca_3SiO_5$ surface.

Understanding the interfacial reactions at the water/$Ca_3SiO_5$ interface using atomistic simulations can provide some new insights on the $Ca_3SiO_5$ dissolution. The adsorption of water on the $Ca_3SiO_5$ surface with molecular and dissociative mode[25] is the first step of $Ca_3SiO_5$ hydration, which happens even before contacting the bulk water due to the strongly hydrophilic nature of $Ca_3SiO_5$[26]. After the surface hydroxylation and the proton hopping into the surface[27], the Ca ion will dissolve into the solution destroying the initial surface topology and promoting the further water penetration[27], which is a key step for advancing the $Ca_3SiO_5$ hydration. For this process, the density functional theory (DFT)-based geometry optimization calculations[28] indicated the adsorption of water on the Ca ion impairs the bond strength

between the calcium and oxygen ions on the surface. Recently, reactive MD simulations have been widely used to study the $Ca_3SiO_5$ dissolution and successfully obtain several new perception on dissolution process. Manzano et al.[27] found the Ca ion desorbs quickly and tends to accumulate as inner- and outer-sphere complexes at the $Ca_3SiO_5$ (111) surface. Qi et al.[29] showed a more easier Ca dissolution from the $Ca_3SiO_5$ (010) surface than the $Ca_2SiO_4$ (100) surface due to the higher surface hydroxylation degree. Sun et al.[30] did not observe dissolution of Ca ions from the (010) surface even after 10 ns at 300 K, but after raising the temperature to ~1000 K, the dissolution rate increases five times than that of room temperature. Claverie et al.[31] first investigated the $Ca_3SiO_5$ hydration using ab initio molecular dynamics (AIMD) simulations and found that the hydroxides formed on superficial oxide ions are highly stable. However, they did not observe an obvious vertical displacement of Ca ions relative to their initial positions. In fact, it is very hard to probe a complete calcium dissolution process at the atomic level using the AIMD simulations[29,32] with small timescale (i.e. within 100 ps). The breakage of Ca–$O_s$ ($O_s$ indicates all the oxygen ion in $Ca_3SiO_5$) bonds and formation of Ca–$O_w$ ($O_w$ indicates the oxygen ion in water) bonds is indeed a rare event for not only AIMD, but also the reactive MD, which calls for the cooperation with the enhancing sampling method, such as metadynamics. Uddin et al.[33] used reactive forcefield (ReaxFF) combined with metadynamics to calculate the free energy changes of dissolution of Ca ions from various $Ca_3SiO_5$ surfaces along the reaction coordinate of the distance between the center of mass and the selected calcium ion. However, this collective variable cannot illustrate the nature of dissolution clearly, which is well accepted as a ligand exchange reaction[34]. Moreover, the chemical reactions at water/$Ca_3SiO_5$ interface are typically accompanied by electron transfer. Hence, it is indispensable to give an ab initio description of such a fundamental reaction.

Here, we uncover the dissolution mechanism of $Ca_3SiO_5$ at early stage with ab initio method. We calculated the reaction pathways, free energy changes and free energy barriers of $Ca_3SiO_5$ dissolution using the ab initio metadynamics simulations. We show that the calcium dissolutions at different sites have different reaction pathways and the less coordinated Ca is easier to escape from the surface. Furthermore, we found that water molecules can reduce the dissolution free energy barriers not only by attractive effect through adsorbing on Ca, but also by repulsive effect through proton penetrating into the surface and water diffusion into the original Ca site. Our findings pave the way to the atomistic understanding of surface reaction for the Ca ion dissolution from $Ca_3SiO_5$.

## Results

**Determination of reaction coordinates and classification of Ca species.** The chemical reaction in initial $Ca_3SiO_5$ hydration, especially the dissolution of Ca ions, is a process of breaking the old Ca–$O_s$ bonds and forming new Ca–$O_w$ bonds. Therefore, we probe into the coordination environment of the Ca ion to calculate the full dissolution pathways. The $Ca_3SiO_5$ dissolution rates at different surface sites (i.e., flat, step and kink site) are typically different due to the different chemical environments around the Ca ion. The coordination environments of the Ca ions on surfaces are various due to the low symmetry of the M3 type of $Ca_3SiO_5$ and the large number of possible surfaces formed during high-temperature calcination process. For example, the Ca coordination environments at seven low-indexes surfaces range from three to seven (Supplementary Table 1). There are four Ca sites in different chemical environments on the $Ca_3SiO_5$ (111) surface and they can be classified into three- and five-

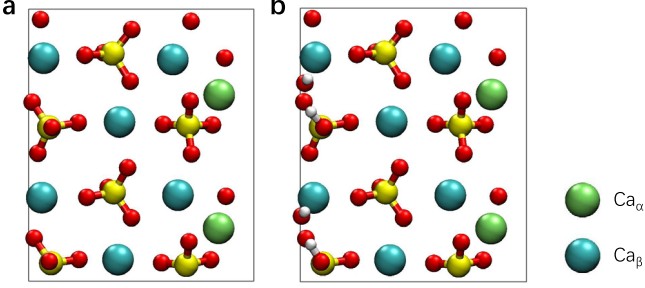

**Fig. 1 The top view of the initial Ca₃SiO₅ (111) surface model. a** The top view of the surface before protonation. **b** The top view of the surface after protonation. The green, cyan, yellow, red, and white spheres are indicted to the three-coordinated $Ca_\alpha$ species, the five-coordinated $Ca_\beta$ species, the silicon, oxygen, and hydrogen ions, respectively.

coordinated Ca species, which are indicated by $Ca_\alpha$ and $Ca_\beta$ in this work, respectively (Fig. 1). Because the surface reactivity varies with the different sites on the $Ca_3SiO_5$ surface and the hydration process is closely correlated with the coordination number of the surface Ca ions[29], we investigate the dissolution mechanism for both $Ca_\alpha$ and $Ca_\beta$.

**Dissolution pathways for $Ca_\alpha$.** For the dissolution of $Ca_\alpha$, we can clearly identify six free energy minima on the two-dimensional (2D) free energy surface (FES) (Fig. 2b). In addition, the free energy basins and free energy barriers along each collective variables (CVs) can also be found through the one-dimensional (1D) FES projected from the 2D FES (Fig. 2a, c). The coordinate of the state is present in form of X(CN(Ca–O_s), CN(Ca–O_w)), where X indicates the state number on the FES. When the water encounters with the $Ca_3SiO_5$ surface, the stable state changes from (3, 0) (the state before water contacting the substrate) to A(3, 2) (Fig. 2d), indicating that two water molecules adsorb on $Ca_\alpha$ and make the system more stable. After crossing two energy barriers ($\Delta A^\ddagger$(A–B) = 3.57 kJ/mol and $\Delta A^\ddagger$(B–C)) = 11.76 kJ/mol), the system comes to the most stable state C(3, 4) with up to four adsorbed water molecules. This high-coordination (seven-coordinated) state compromises the breakage of the original Ca–O_s bond but associated with huge free energy barriers ($\Delta A^\ddagger$(C–D)) = 15.71 kJ/mol and $\Delta A^\ddagger$(D–E)) = 14.28 kJ/mol) and a little increase in free energy changes ($\Delta A$(C–D = 4.69 kJ/mol and $\Delta A$(D–E = 3.48 kJ/mol)). The breakage of the bond between Ca and O_si (the oxygen ion from the silicate group in $Ca_3SiO_5$) is earlier than that between Ca and O_i (the interstitial oxygen ion in $Ca_3SiO_5$) due to the sequence of reaction pathways from state C to D to E. While the breakage of Ca–O_si bond is more difficult than that of Ca–O_i bond owing to the higher free energy barriers between states C and D. These two sequential steps of breaking Ca–O_s bonds decrease the total coordination number from seven to five, making this detached and free Ca ion have more chances to accommodate one more water ligand and reform an octahedral structure, although at this stage it is severely distorted with a trigonal bipyramid ($D_{3h}$) structure. The free energy barrier and the free energy change for these steps ($\Delta A^\ddagger$(E–F) = 8.70 kJ/mol and $\Delta A$(E–F) = −3.87 kJ/mol) are relatively high compared to the same fivefold to sixfold coordination transition step (A–B).

**Dissolution pathways for $Ca_\beta$ with CN(Ca–O_s) from 5 to 2.** The coordination environments of Ca ions may change the dissolution pathways as well as the thermodynamic and kinetic properties. Thus, we carried out the comparative WT-MetaD simulations for $Ca_\beta$ to investigate whether the dissolution

pathway alters with initial coordination environment. Obviously, the FES for the detachment of $Ca_\beta$ is different from that for $Ca_\alpha$ (Fig. 3a–c) and becomes more complicated with more possible reaction pathways. When water molecules come to the $Ca_3SiO_5$ surface, the first stable state is A(5, 1). Albeit the number of water molecule is less than that for $Ca_\alpha$, the total coordination number is same with six. However, the reaction pathway for the next step is more complex. The state A has two potential reaction paths to adsorb more water molecules. The first one adsorbs one more water molecule directly without breaking the Ca–O_s bond (A–B). While the other one does it by breaking two Ca–O_s bonds at the same time (A–D). From a thermodynamic point of view, it is more energetically favorable to pass through state B first due to the larger free energy change between the state A and the next free energy minimum. Nevertheless, from a kinetic point of view, it is faster to react along the pathway of A–D because of its lower free energy barriers ($\Delta A^\ddagger$(A–D) = 6.49 kJ/mol and $\Delta A^\ddagger$(A–B) = 7.01 kJ/mol). The reaction pathway of A–B–C–D for the dissolution of $Ca_\beta$ is similar to the B–C–D–E for the dissolution of $Ca_\alpha$, in which the start of Ca–O_s bond cleavage is from the sevenfold coordination state. Noticeably, the seven-coordinated species is an essential intermediate in the dissolution of Ca, which is similar to the decomposition of $\gamma$-$Al_2O_3$[35]. But the free energy barriers along the B–C–D ($\Delta A^\ddagger$(B–C) = 7.38 kJ/mol, $\Delta A^\ddagger$(C–D) = 4.24 kJ/mol) is much smaller than that along C–D–E for $Ca_\alpha$. The state D(3, 2) is the most stable state with the same coordinate of the start point, A, on the FES of $Ca_\alpha$ and is also a new outset for other two different reaction pathways towards breaking one more Ca–O_s bond. The reaction would be more likely to proceed in the direction from D to E(2, 2) due to the lower free energy barrier ($\Delta A^\ddagger$(D–E) = 19.60 kJ/mol) compared to the route from D to G(2, 3) ($\Delta A^\ddagger$(D–E) = 29.02 kJ/mol, $\Delta A^\ddagger$(E–F) = 26.11 kJ/mol)) and it is the rate-controlling step among all the reactions. It should be noted that the further Ca–O_s bond cleavage from 2 to 0 is not accessible during this simulation due to the large free energy barriers required. Therefore, to uncover the subsequent dissolution mechanism of Ca with less than two- and even zero-coordinated O_s, it is necessary to add a 'wall' to constrain the CVs in the region of interest.

**Complete dissolution of $Ca_\beta$ with CN(Ca–O_s) from 2 to 0.** For the further detachment of $Ca_\beta$ with CN(Ca–O_s) from 2 to 0, we assume that the start point is the most stable state H(1, 4) on the FES (Fig. 4a) as it is the most likely to exist in practice. To further dissolve, $Ca_\beta$ needs to guest a water molecule first to achieve an octahedral structure crossing over a 26.99 kJ/mol free energy barrier and coming to the state I(1, 5) (Fig. 4b). The next step of associating one more water molecule from the state I to K(1, 6) is the rate-controlling step due to the highest free energy barrier of 28.08 kJ/mol. After that, $Ca_\beta$ detaches from the original position progressively and finally gets rid of the confinement of O_s network totally, which is hydrated by the surrounding water molecules to the six- or seven-coordinated solute ion. The five-, six- and seven-coordinated $Ca_\beta$ ion can be transformed to each other. However, the sixfold coordination state is the most stable, not only because the reactions from the fivefold state M(0, 5) and sevenfold state L(0, 7) to the sixfold state K(0, 6) are spontaneous ($\Delta A$(M–K) = −2.09 kJ/mol and $\Delta A$(L–K) = −10.30 kJ/mol), but also because the free energy barriers ($\Delta A^\ddagger$(M–K) = 10.34 kJ/mol, $\Delta A^\ddagger$(L–K) = 6.96 kJ/mol) are lower than those of the reverse reactions, which is confirmed by an additional 30 ps equilibrium AIMD simulations. This final run also shows that the CN(Ca–O_w) increased slightly, and the dissolved Ca forms a more regular octahedral structure with water and hydroxyl with the time evolution (Supplementary Movie 1).

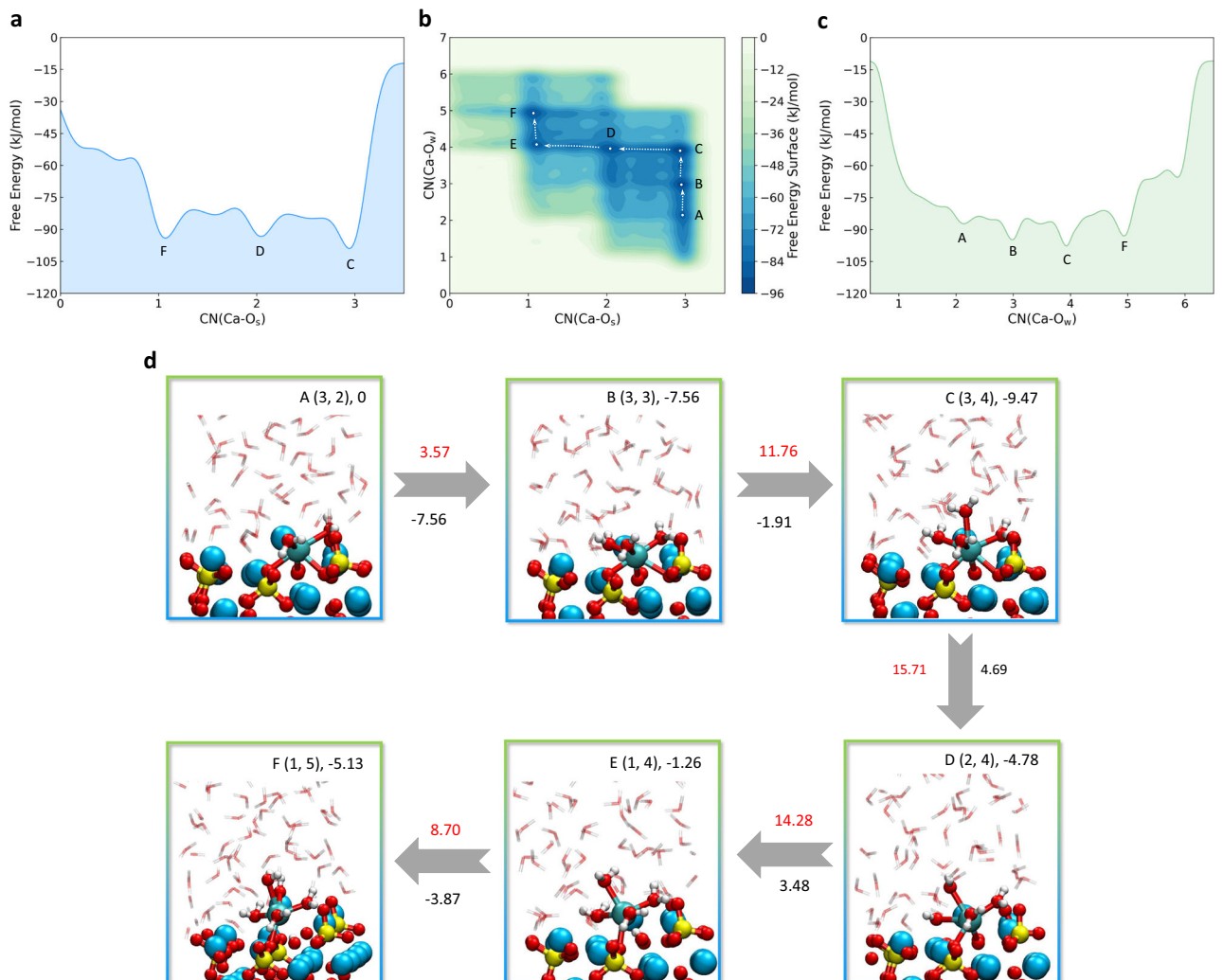

**Fig. 2 Dissolution mechanism of Ca$_\alpha$ from the Ca$_3$SiO$_5$ surface. a**, **c** The one-dimensional free energy profiles with respect to CN(Ca–O$_s$) and CN(Ca–O$_w$), respectively. **b** The two-dimensional free energy surface with variables of CN(Ca–O$_s$) and CN(Ca–O$_w$). **d** The configurations of the free energy minimum states on the FES and the corresponding reaction pathways. The state number, coordinates on the FES and the Helmholtz free energy values (kJ/mol) relative to state A are at the upper right. The values in red are free energy barriers (kJ/mol) and the values under the arrows in black are overall changes in free energies between two states (kJ/mol). The yellow, blue, cyan, red, and white spheres are indicted to the silicon, calcium with no bias potential, calcium with bias potential, oxygen and hydrogen ions, respectively. For simplicity, the solute is shown in the transparent stick type.

**Structural and spectroscopy analyses of the water/Ca$_3$SiO$_5$ interface after the dissolution of Ca.** According to the atomic density and atomic excess profiles (Fig. 5a), we define six regions for the water structure along the $z$ direction on the Ca$_3$SiO$_5$ surface. In the region I (6–8.7 Å), it is obvious that the H ion penetrates into the second layer of the surface ($z = 6$ Å), due to the escape of the Ca ion and the proton exchange from water molecules to the inner O$_i$ ion. Additionally, the position of the peak of O$_w$ and H is overlapped at 8 Å, indicating one water molecule settles down the original position of the dissolved Ca ion with its dipole moment parallel to the surface. The region II (8.7–11.1 Å) is the chemisorbed water molecule region showing an apparent and strong peak for both O$_w$ and H at nearly the same position, which means the chemisorbed water molecules have a well-ordered structure and tend to orient their dipole moments parallel to the surface. This result is different from our previous work on the Ca$_3$SiO$_5$ surface without Ca dissolution[25], which presents a tendency for upright configuration with O$_w$ below H. This divergence indicates the dissolved Ca ion tunes the direction of the chemisorbed water to a flatter configuration. The

region III (11.1–14.3 Å) is a mixture of the physisorbed water molecule for the surface and the first hydration shell of the dissolved Ca ion. In this region, the obvious overlap of the peaks for O$_w$ and H disappears, indicating a destruction of the layered water structure. The region IV (14.3–20.8 Å) is the transition layer with a less extent of the structuration and ae oscillating at 0. The region V (20.8–28 Å) is the bulk water layer with ae almost 0, and region VI (28–33 Å) is the water/vacuum layer. The regions of IV, V, and VI are similar to those on the Ca$_3$SiO$_5$ surface without Ca dissolution not only in width but also in intensity. We also calculated the infrared (IR) spectra for the system and extract the parts for Si–OH and Ca–OH. It clearly shows one band at 916 cm$^{-1}$ (Fig. 5b), which is characteristic for the Si–OH bond raised in experimental IR spectra upon Ca$_3$SiO$_5$ hydration[19]. In addition, two bands arising at 700–1000 cm$^{-1}$ and 3640 cm$^{-1}$ (Fig. 5c) shows the formation of Ca–OH during the dissolution of Ca as indicated by the experimental results[19,36]. The radial distribution function (RDF) (Fig. 5d) shows that the intensity of the peaks which corresponds to the first and second hydration shells of the dissolved Ca ion is greater than of the surface Ca ion. It also

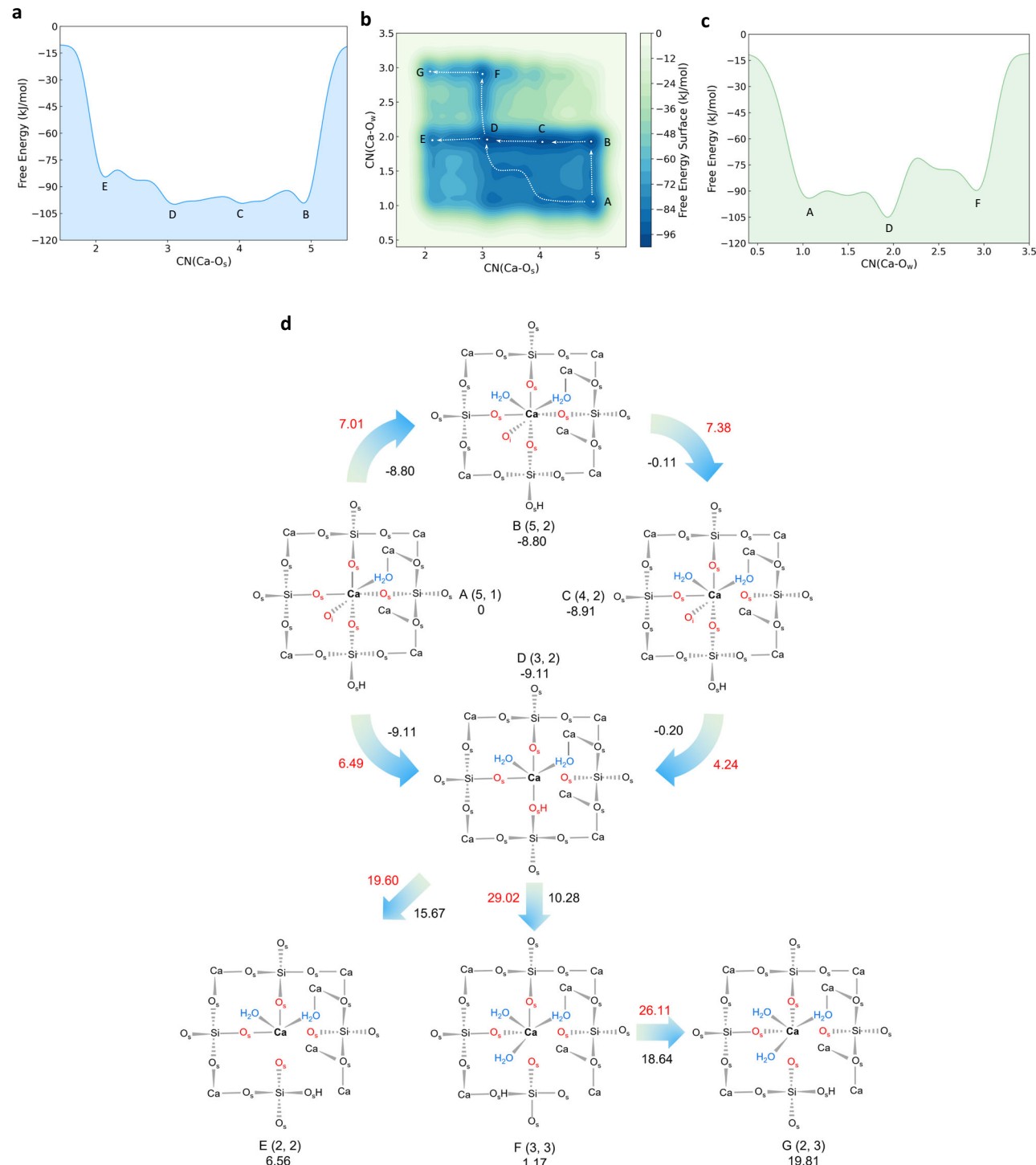

**Fig. 3 Dissolution mechanism of $Ca_\beta$ from the $Ca_3SiO_5$ surface with $CN(Ca–O_s)$ from 5 to 2. a, c** The one-dimensional free energy profiles with respect to $CN(Ca–O_s)$ and $CN(Ca–O_w)$, respectively. **b** The two-dimensional free energy surface with variables of $CN(Ca–O_s)$ and $CN(Ca–O_w)$. **d** The overhead sketches for the first layer of the $Ca_3SiO_5$ surface of the free energy minimum states on the FES (the all-atom configurations is presented in Supplementary Fig. 1) and the corresponding reaction pathways. The state number, coordinates on the FES and the Helmholtz free energy values (kJ/mol) relative to state A are presented around the corresponding structure. The values in red are free energy barriers (kJ/mol) and the values under the arrows in black are overall changes in free energies between two states (kJ/mol). The biased $Ca_\beta$ is bolded, and the $O_s$ and $O_w$ bonded with $Ca_\beta$ were highlighted in red and blue, respectively.

presents that the Ca–$O_w$ bond length for the dissolved Ca ion is 2.39 Å, which is shorter than that for the surface Ca ion with 2.50 Å, indicating a stronger interaction between the Ca ion and water molecules after dissolving. From the final snapshot of the equilibrium AIMD simulations (Fig. 5e), the $Ca_3SiO_5$ dissolution

can be interpreted as a process where the Ca ion coordinated with five water molecules and one hydroxyl group releases from the $Ca_3SiO_5$ surface with the proton transferring from the water molecule to the second layer $O_i$ ion, and the occupation of the initial Ca site by one water molecule. This water molecule is

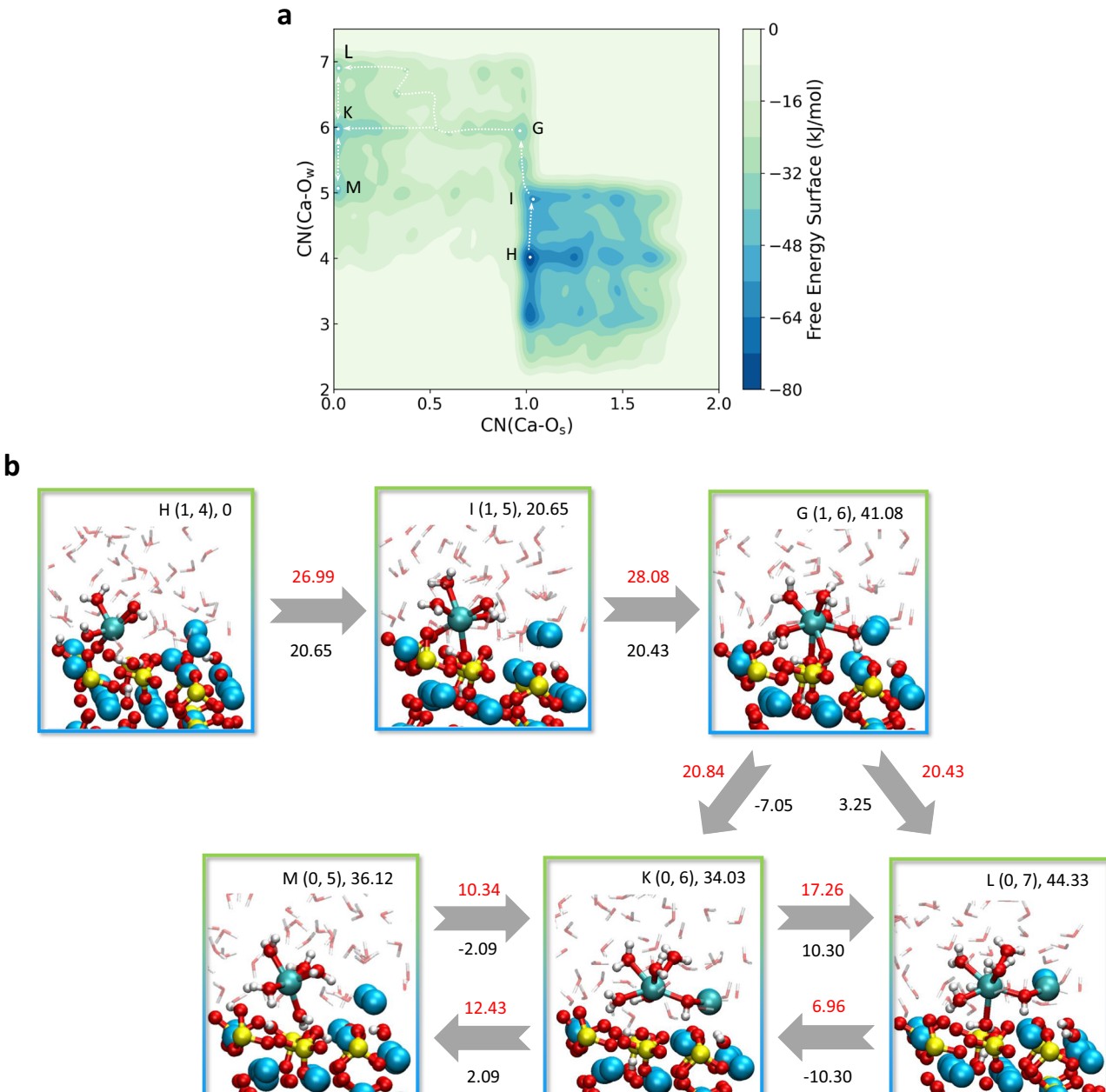

**Fig. 4 The further dissolution mechanism of Ca$_\beta$ from the Ca$_3$SiO$_5$ surface with CN(Ca–O$_s$) from 2 to 0. a** The two-dimensional free energy surface with variables of CN(Ca–O$_s$) and CN(Ca–O$_w$). **b** The configurations of the free energy minimum states on the FES and the corresponding reaction pathways. The state number, coordinates on the FES and the Helmholtz free energy values (kJ/mol) relative to state H are at the upper right. The values in red are free energy barriers (kJ/mol) and the values under the arrows in black are overall changes in free energies between two states (kJ/mol). The yellow, blue, cyan, red, and white spheres are indicted to the silicon, calcium with no bias potential, calcium with bias potential, oxygen and hydrogen ions, respectively. The calcium with no bias potential but connecting with Ca$_\beta$ through hydroxyl group is also shown in cyan. For simplicity, the solute is shown in the transparent stick type.

parallel to the surface and forms hydrogen bonds with the newly formed hydroxyl group and O$_s$ ions.

## Discussion

The calcium sites with different coordination environments lead to different reaction pathways, free energy barriers, and free energy changes. The dissolution of three-coordinated Ca$_\alpha$ is easier than the five-coordinated Ca$_\beta$ not only because of its initial less restraint from the Ca$_3$SiO$_5$ surface, but also the smaller free energy barriers along the reaction pathways. In addition, The free energy barriers between the two stable states on the FES of either Ca$_\alpha$ or

Ca$_\beta$ tends to be larger as the number of Ca–O$_s$ bonds decreases, which means the water adsorption on the Ca$_3$SiO$_5$ surface is easier than the detachment of Ca and the kinetic rate decreases gradually as this process proceeds. Nonetheless, even the highest free energy barrier is only 29 kJ/mol, which is easy to be crossed, suggesting the Ca dissolution is an auto-catalytic process. Besides, the free energy changes for the detachment of Ca$_\alpha$ is negative, while for the detachment of Ca$_\beta$ is positive, which indicates that the dissolution of Ca$_\alpha$ is spontaneous and Ca$_\beta$ is unspontaneous. However, Uddin et al.[33] pointed out that the reactivity of different surfaces for Ca dissolution would be different, thus the value of free energy

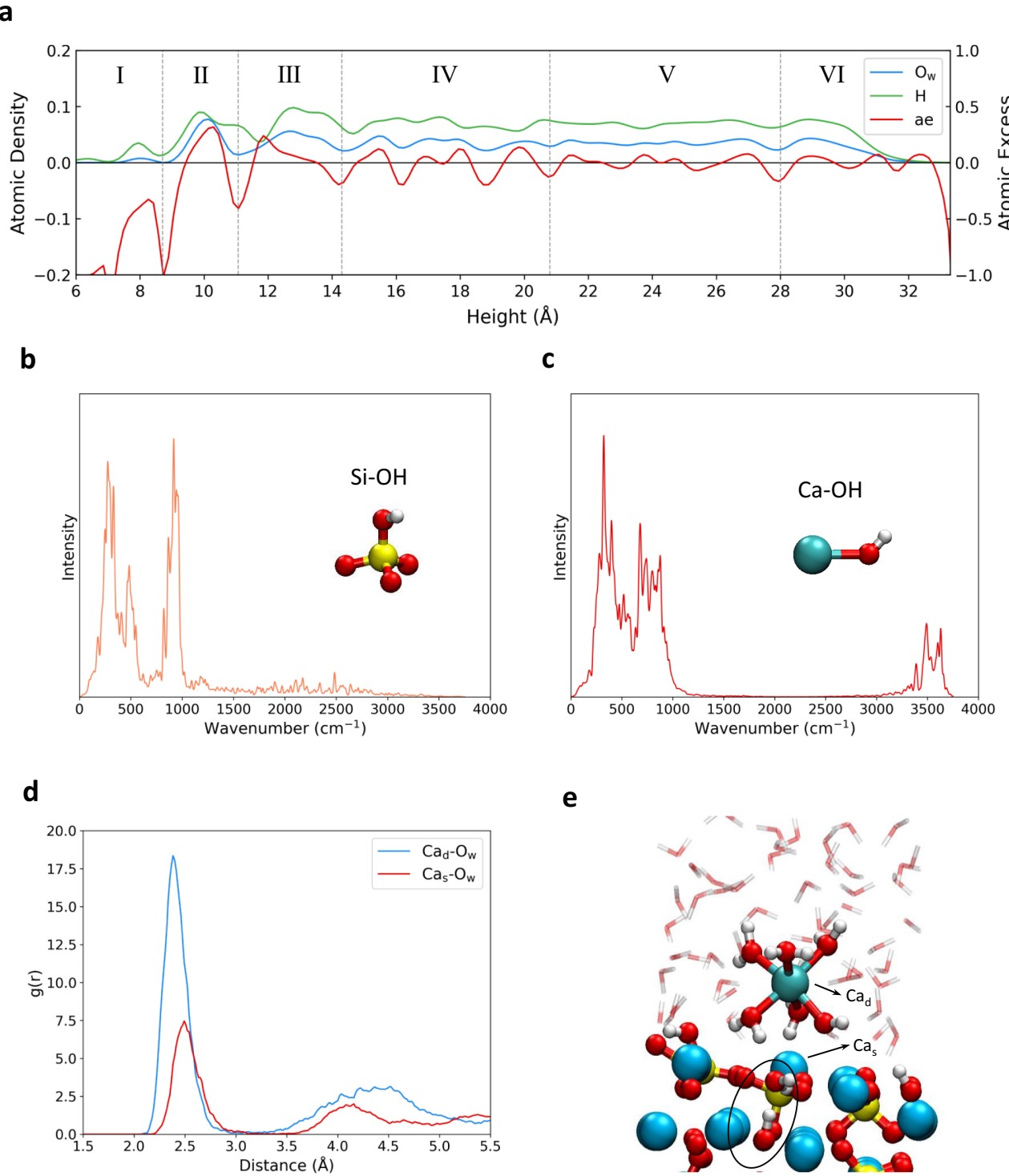

**Fig. 5 The structural and spectroscopy analysis of the water/Ca₃SiO₅ interface after the dissolution of Ca. a** Atomic density for $O_w$ and H, and atomic excess profiles as a function of the height beginning at 6 Å from the bottom of the $Ca_3SiO_5$ surface. **b**, **c** The calculated IR spectrums for Si–OH group and Ca–OH group, respectively. **d** Radial distribution function (RDF) between the dissolved Ca ion ($Ca_d$) and the $O_w$ ion as well as between the surface Ca ion in the same site ($Ca_s$) and the $O_w$ ion. **e** The snapshot of the equilibrium AIMD after 30 ps.

changes and barriers would vary with Miller indices. It should be noted that there is no human intervention during the simulation of crossing over the free energy barriers, which ensures the reliability of our results.

Our ab initio WT-MetaD simulations with explicit water solvation highlight the importance of the water molecules on the detachment of Ca during the $Ca_3SiO_5$ hydration. In short, the dissolution of Ca can be explained in terms of a ligand exchange process. It initially stimulated by the water adsorption, raising the total coordination number with oxygen ions to a high level (five to seven). The adsorbed water molecules reduce the free energy barriers for breaking the $Ca–O_s$ bond, and thus provide an opportunity for Ca to break the $Ca–O_s$ bonds with the optimal six coordination number unchanged. In fact, the breakage of $Ca–O_s$ bonds is a multi-step

and multi-orientation chemical reaction, and every step needs a relatively high free energy barrier, which is strait to cross for traditional AIMD simulations within 100 ps but easy in practice. In addition, the dissolution of Ca is further promoted by the proton exchange and the diffusion of the water molecule from the chemisorbed layer into the second surface layer. On the one hand, the H penetrates into the second layer of the surface and bonds to the $O_i$ previously bonded to the dissolved Ca, leading to a repulsive force pushing Ca out of the surface. On the other hand, this diffused water molecule resides in the position of the Ca before dissolution and forms the hydrogen bond network with $O_s$, $O_w$, and H, which undermines the attractive force to this Ca ion.

In summary, an atomistic and mechanistic picture of the Ca dissolution from the $Ca_3SiO_5$ surface in water solution at the initial stage of $Ca_3SiO_5$ hydration is investigated using the ab initio molecular dynamics and metadynamics simulations. We find that the Ca dissolution from the $Ca_3SiO_5$ are multi-step and multi-orientation chemical reactions accompanied by the water adsorption, proton exchange, breakage of $Ca–O_s$ bonds, and water diffusion. The reaction pathways for Ca in different coordination environments are different and the less coordinated Ca is easier to leach from the surface. The thermodynamic and kinetic analyses show that the detachment of $Ca_\alpha$ is spontaneous, while the detachment of $Ca_\beta$ is unspontaneous, and the Ca dissolution is an auto-catalytic process with the highest free energy barriers of only 29 kJ/mol. Besides, we find the water molecules provide not only the attractive force pulling Ca out of the surface, but also the repulsive force in filling the previous Ca site and pushing Ca away. The present achievement thus provides an insight into the cement hydration and also predict the evolution of other complex geochemical and catalytic systems.

## Methods

**Atomistic model.** Because the $Ca_3SiO_5$ (111) surface has been studied to have a greater possibility to form in practice[25,37], we took the pure $Ca_3SiO_5$ (111) surface from our previous work[25], which was cleaved from the M3 polymorph of $Ca_3SiO_5$ (obtained from CCSD[38]), as the M3 polymorph is the most frequently observed in industrial clinkers together with the M1 form[39]. The details of the DFT-based geometry optimization of the bulk crystal and surface slab were listed in Supplementary Methods. Considering the adsorption of water molecules on the $Ca_3SiO_5$ surface occurs even before contacting bulk water[40], we firstly adsorbed isolated water molecules to saturate the dangling bond on the $Ca_3SiO_5$ surface (Fig. 1b). Then, we put a 20 Å thick layer of water with density of 1 g/cm³ on the $Ca_3SiO_5$ (111) surface (totally 333 atoms) for the dissolution simulations. The lattice parameters were 14.21 Å × 11.72 Å × 36 Å after setting a vacuum of 15 Å along z direction. After the detachment of the $Ca_\beta$ from its initial position to the state of D, we further constructed a new model (14.21 Å × 11.72 Å × 48 Å) by adding another 10 Å thick layer of water on the previous system (totally 498 atoms) to calculate the structural and dynamic properties of the equilibrium state.

**AIMD simulations.** All the AIMD simulations reported in this work were performed within the framework of DFT with the generalized gradient approximation (GGA) using the Perdew-Burke-Ernzerhof (PBE)[41] functional and Grimme D3 correction[42], which was implemented in the CP2K/Quickstep code[43]. The Core electrons were described by Goedecker-Teter-Hutter (GTH) pseudopotentials[44,45] and the valence electrons were described by a mixed Gaussian and plane waves basis (GPW)[46]. The wave functions were expanded on a double-ζ valence polarized (DZVP) basis set along with an auxiliary plane wave basis set at a cutoff energy of 500 Ry. The Brillouin zone was sampled by the gamma approximation. During AIMD, the nuclei were treated within the Born–Oppenheimer approximation with a timestep of 0.5 fs for equilibrium simulation, while 1 fs for metadynamics simulations with the replacement of hydrogen by deuterium to accelerate the structural evolution without energy drifts[31,47]. The temperature was maintained at 300 K using a Nosé-Hoover thermostat[48,49] coupled to the system with a time constant of 1000 fs in the Canonical ensemble (NVT). The convergence criterion for energy was set to $10^{-12}$ Hartree and for self-consistent field was $10^{-6}$ Hartree. All the system were first optimized to a stable state and then thermalized for at least 2.5 ps before the production run for statistical analysis. The production times for different simulations are shown in Table 1.

**Table 1 The production times for all the simulation projects.**

| Simulation project | Production time |
| --- | --- |
| Dissolution of $Ca_\alpha$ | 100 ps |
| Dissolution of $Ca_\beta$ with CN(Ca-$O_s$) from 5 to 2 | 63 ps |
| Further dissolution of $Ca_\beta$ with CN (Ca-$O_s$) from 2 to 0 | 46 ps |
| Equilibrium of final state of $Ca_\beta$ | 30 ps |

**Metadynamics simulations.** In the well-tempered metadynamics[50] (WT-MetaD) simulations, we utilized a two-dimensional collective variables (CVs) characterized by the coordination number (CN) to monitor the dissolution process. The CN(Ca–$O_s$) is the coordination number of the Ca ion with all oxygen ions from the surface slab, while CN(Ca–$O_w$) is the coordination number of the Ca ion with all oxygen ions from water molecules. As defined in the PLUMED code[51], the CN have the expression as follows:

$$CN(Ca, O_{w/s}) = \sum_{j \in O_{w/s}} s_{ij}(r_{ij}) = \sum_{j \in O_{w/s}} \frac{1 - \left(\frac{r_{ij}-d_0}{r_0}\right)^n}{1 - \left(\frac{r_{ij}-d_0}{r_0}\right)^m} \quad (1)$$

where $r_{ij}$ is the distance between atom i and atom j. $s_{ij}(r_{ij})$ is a rational type of switching function describing the coordination between atom i and j. $d_0$ is the central value of the function. $r_0$ is the acceptance distance of the switching function, where the function well be n/m at $d_0 + r_0$. Here, we define $d_0$ is 2.42 Å, which is the equilibrium bond length between the Ca and O ions[52]; $r_0$ is 0.4 Å, which is around half of the full width at half maximum of the radial distribution function of Ca-O[53] and n and m are 6 and 12, respectively.

The Gaussian hills were deposited every 30 timesteps with the initial height of 3.5 kJ/mol and width of 0.15 for both CVs. The biasfactor were 15 for simulations of $Ca_\alpha$ and 24 for simulation of $Ca_\beta$. In addition, for further investigation of $Ca_\beta$ coordinating more water molecules, we added a quadratic wall with the force constant of 500 kJ/mol at the position of CN(Ca–$O_s$) equals to 1.5 to restrict the simulation of further dissolution of $Ca_\beta$ on the regions of free energy surface with CN(Ca–$O_s$) less than 1.5. The time evolutions of the CV1 and CV2, the convergence tests for the free energy surfaces and the errors between the free energy minima were shown in Supplementary Figs. 2–7.

**Structural and spectroscopy calculations.** The atomic excess (ae) is defined as:

$$ae = \frac{2[O_w] - 2[H]}{2[O_w] + 2[H]} \quad (2)$$

where $[O_w]$ and [H] are atomic density for $O_w$ and H, respectively. The negative value for ae indicates an excess of H, while the positive one indicates an excess of $O_w$.

For the vibrational spectra, we use the last 30 ps equilibrium AIMD trajectory to calculate the infrared (IR) spectrum with the TRAVIS program[51]. The molecular electric properties were calculated every 4 fs (8 timesteps) using the Voronoi integration approach[54]. The IR spectrum of particular components of a system were computed through the Fourier transform of the molecular dipole autocorrelation function as follows:

$$A(\omega) \propto \int \langle \dot{\mu}(\tau)\dot{\mu}(t+\tau)\rangle_\tau e^{-i\omega t} dt \quad (3)$$

where $A$ is the absorption cross section, $\omega$ is the frequency, and $\dot{\mu}$ is the time derivative of the dipole moment leading to the dipole-velocity autocorrelation function.

## Data availability

The data supporting this study are available from the corresponding author upon reasonable request.

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

## Acknowledgements

The authors gratefully acknowledge the financial support provided by The Science and Technology Development Fund (FDCT), Macau (0138/2020/A3) [Li] and the Hong Kong Research Grants Council (T22-502/18-R) [Li].

## Author contributions

Z.L. and H.P. conceived and supervised the project. Z.L., Y.L., and H.P. designed the research. Y.L. performed the simulations, analyzed the simulation results and wrote the manuscript. Q.L. and X.M. discussed the results. Z.L., Y.L., and H.P. edited the manuscript before submission.

## Competing interests

The authors declare no competing interests.

**Additional information**

