## [Peer Review File · Nature Communications]

REVIEWER COMMENTS

Reviewer #1 (Remarks to the Author):

The paper “NCOMMS-21-44726, Ab initio mechanism revealing for tricalcium silicate dissolution” starts with an interesting premise: compute the dissolution energy barriers of C3S using atomistic simulation methods. However, soon in the introduction I have the feeling that the authors are over-selling their research. For instance, starting in line 58:

“All these hypotheses and the fit of the calorimetric curve of Ca₃SiO₅ hydration using thermodynamic calculations may be derived from the ignorance on the interfacial reactions during the Ca₃SiO₅ dissolution, especially the dissolution behavior of calcium ions at atomic level. Fortunately, atomistic simulations can tackle these problems.”

Atomistic simulation results, including those presented in this paper” are truly far from explaining the calorimetric curve, way further than the “thermodynamic calculations may be derived from the ignorance” that the authors mention. AS a matter of fact, we don't even know if dissolution is controlling the hydration of C3S, or the nucleation of C-S-H is the main mechanism. In addition, they mention “especially the dissolution behavior of calcium ions at atomic level”. Why Ca ions? Why not SiO₄ ions? there is not reason behind this statement.

All the text starting in line 65:

“Claverie et al.²⁷ investigated the proton transfer at the water/Ca₃SiO₅ interface using ab initio molecular dynamics (AIMD) simulations and found that the hydroxides formed on the surface are highly stable. However, they did not observe an obvious vertical displacement of Ca ions relative to the initial position. In fact, it is very hard to probe a complete calcium dissolution process at the atomic level even using the traditional molecular dynamics (MD) simulations^{28, 29} with large timescale (i.e. nanoseconds). Moreover, the classical MD is not appropriate to simulate the chemical reaction involving the breakage and formation of bonds. Even using the ReaxFF force field is still not preciously enough to present the reaction pathways for a chemical reaction.”

Is written to justify the present work, and has no fundament. Cleaverie et al. did not observe Ca dissolution, true, and traditional MD cannot reproduce water dissociation, but then, Why “the ReaxFF force field is still not preciously enough to present the reaction pathways for a chemical reaction”? There is no reasoning or citation behind this statement. In fact, ReaxFF simulations have been the most useful simulation studies so far. They are able to reproduce the dissolution of Ca atoms (see - Manzano, H., Durgun, E., López-Arbeloa, I., & Grossman, J. C. (2015). Insight on tricalcium silicate hydration and dissolution mechanism from molecular simulations. ACS applied

materials & interfaces, 7(27), 14726-14733, Qi, C., Manzano, H., Spagnoli, D., Chen, Q., & Fourie, A. (2021). Initial hydration process of calcium silicates in Portland cement: A comprehensive comparison from molecular dynamics simulations. *Cement and Concrete Research*, 149, 106576. and Salah Uddin, K. M., & Middendorf, B. (2019). Reactivity of Different Crystalline Surfaces of C3S During Early Hydration by the Atomistic Approach. *Materials*, 12(9), 1514.) and the accuracy has not been any discussion about the precision of the method. Why is ReaxFF “not precisely enough”?

The authors present previous work in the field, but they do not mention several very relevant papers, specially:

- Manzano, H., Durgun, E., López-Arbeloa, I., & Grossman, J. C. (2015). Insight on tricalcium silicate hydration and dissolution mechanism from molecular simulations. *ACS applied materials & interfaces*, 7(27), 14726-14733.

- Salah Uddin, K. M., & Middendorf, B. (2019). Reactivity of Different Crystalline Surfaces of C3S During Early Hydration by the Atomistic Approach. *Materials*, 12(9), 1514.

The second one is specially relevant, as they use metadynamics to compute dissolution energy barriers. They use ReaxFF instead DFT, but then the methodology is more or less the same as in the present paper. The results are very different, and even if I believe that the present work is more rigorous, the results should be compared and discussed.

The simulations of the free energy are interesting, nicely done and nicely presented. As I said, I think that this work is better done than Uddin et al. But I think that there is a mayor methodological problem: why the 111 surface? It is hard to say, but I would say given the structure and symmetry of the MIII polymorph that the created slab is not symmetric. Therefore, how is the plane chosen? Did the authors do any surface reconstruction? are Ca atoms symmetrically distributed in the top and bottom of the slab? The results are compromised by these factors.

There are several surprising statements when the results are tried to be compared with experiments. For instance: “The mean square displacement (MSD) (Figure 3c) presents a more dynamic property and stronger diffusion ability for the dissolved Ca compared to the previous surface state” Well, of course the atoms will displace more in solution than in the solid, the result has no relevance. As the IR spectra: if there are Si-OH and Ca-OH in the system, the IR spectra will match the experiment, no need to do it. But more surprisingly is the comparison in figure 4f: “From the transmission electron microscopy (TEM) photograph (figure 4f), we can clear see when Ca₃SiO₅ encounters with water, the Ca cations are released from the Ca₃SiO₅ surface and form the Ca(OH)₂ in the aqueous solution.” In the TEM image that cannot be seen. Atoms, or complexes, cannot be

seen. Those “particles” are way larger than $\text{Ca}(\text{OH})_2$, and they might be more likely C-S-H particles precipitated at the surface or less dense C3S region due to partial dissolution.

I was extremely surprised when I arrived to line 308 and I realise that TEM is from the authors, not from another work. I don't understand what the TEM is contributing to the paper^[1]_{SEP}.

Overall, I recommend rejection of the paper. The calculation of the energy barriers are in my opinion interesting, but the paper over shells the relevance. They do not cite all the relevant papers, they make statements that are clearly wrong, and there are methodological problems.

Reviewer #2 (Remarks to the Author):

The manuscript NCOMMS-21-44726 « Ab initio mechanism revealing for tricalcium silicate dissolution » presents atomic scale simulations of calcium cations dissolution at the (111) surface of $\text{M}_3\text{Ca}_3\text{SiO}_5$ (C3S in cement chemistry notation). The authors ran well-tempered ab initio metadynamics simulations to determine reaction pathways, free energy changes and free energy barriers. The results of this study are originals and of importance for the fundamental understanding of Ca_3SiO_5 dissolution.

In spite of the originality and the considerable work needed for simulations and analysis, several points should be considered to improve the manuscript:

1) It is not indicated to which oxygen atoms O_s refers to. Generally, O_s refers to oxygen in silicates, but I supposed that here O_s refers to superficial oxygen atoms without discrimination between isolated oxygen anions (O_i) and oxygen in silicates. In both cases, the naming code for chemical species (O_w , O_s) should be provided, and more information should be given on Ca- O_i bonds for which the breakage occurs earlier (l. 106-108 for ex.).

2) a) It is also not clear why the (111) surface was chosen for the study. To facilitate the reader's understanding, a top view of this surface before and after protonation occurs, indicating the position of Ca_α and Ca_β , should be provided. It would be also pertinent to discuss on the difference in the environment of Ca at this surface compared with other ones.

b) Furthermore, the statement: “the results obtained in this work are general and applicable to other kinds of calcium silicate species” (l. 209-210) should be revised. In my opinion, the results are very specific to C3S and arguably to the surface under study. If not so, the applicability of the results to other minerals must be proven. The method however can be applied to other minerals.

c) The formation of stable hydroxyl on the surface before starting the production runs should be discussed because this may influence the dissociation of Ca²⁺. Moreover, it is not stated when the proton transfer to the Oi of the second layer occurs, and when the water molecule comes to occupy the calcium site. A timeline with the important events could improve the quality of the manuscript.

3) According to l. 176-177, the regions IV, V and VI are similar to the “perfect” Ca₃SiO₅. Are they similar in width, in intensity and/or position of the peaks? What about the other regions (I, II, III) ?

4) More details should be provided with regard to the following sentence on the spontaneous and unspontaneous nature of Ca dissociation: “The thermodynamic and kinetic analyses show that the detachment of Ca is spontaneous when the Ca is not fully dissolved and unspontaneous when Ca is no more coordinated with Os.” (l. 231-233).

5) Other small changes may improve the quality of the manuscript:

- The duration of the production and equilibrium runs should be given in a list or a table to improve the readability (l. 272-275)

- “the M3 type of Ca₃SiO₅ (obtained from CCSD33), which is most frequently observed in industrial clinkers” → “the M3 polymorph of Ca₃SiO₅ (obtained from CCSD33), which is the most frequently observed in industrial clinkers together with the M1 form”

- The region of the zoom in Fig 4f should be reduced accordingly to the scale bar. The snapshot of the simulation is ~1 nm in width.

- Metadynamics simulations applied to calcium dissociation from Ca₃SiO₅ surfaces was already performed with ReaxFF. A reference to doi:10.3390/ma12091514 would be pertinent.

6) The English language of the manuscript is generally understandable but needs an in-depth revision to improve the quality of the manuscript. E.g. “lead to that” (l. 16); “preciously enough” (l. 72); “any one of” (l. 97) → “each”; “indicating two water...” (l. 100) → “indicating that two water”; “rapider” (l.130) → “quicker” or “faster”; “possible” (l.139; 150) → “likely”; “but also” (l. 160) → “but also because”?; “a further 30 ps equilibrium AIMD simulations” (l.162) → “an additional 30 ps equilibrium AIMD simulation”; “The AIMD also shows that the coordinated Ow with Caβ” (l.162-163) → “This final run also shows that CN(Ca-Ow)”; “evolution of time” (l. 164) → time evolution; “in the

hydration of Ca₃SiO₅ to be observed by the AIMD simulation" (l. 172) → "of the hydration of Ca₃SiO₅ to be observed by AIMD simulation"; "The region II is the chemisorbed water molecule and III is ... water molecule for the surface" (l. 172-173) → "Water in the region II is chemisorbed and the region III corresponds to ... water molecule on the surface"; "new formed" (l.196) → "newly formed"; "Ca-Os bonds become less" (l.204) → "number of Ca-Os bonds decrease"; "hydron" → "hydrogen" (l. 225); "precious" (l. 238) → "previous"; l. 260 "the CP2K/Quickstep" → "the CP2K/Quickstep code"; "the Gaussian hills was" (l.289) → "the Gaussian hills were"; "though" → "through" (l.304), " ω is frequency" (l. 306) → " ω is the frequency"

Some sentences are hard to understand and should be rephrased/corrected:

- l. 103-106: (e.g. "increase in free energy" should be more appropriate than "increase in free energy changes")

l. 106-108: (e.g "than that with" → "than with")

l. 123-124: "Albeit" (l. 123) → "Although"

l. 128-129: Be more specific about the "second free energy minimum". "pass through state B first". Why "first"?

l. 150: "to exist in reality", what is meant by "in reality"?

l. 152-153 "welcoming one more water molecule"

l. 158: "processes the greatest possibility"

l. 178-179: "the first and second hydration shells ... are more than" → "the intensity of the peaks which corresponds to the first and second hydration shells...is greater than"

l. 181-183: "a more dynamic property and stronger diffusion ability"

l. 190-196: the sentence is long and hard to understand in some parts (e.g. "the proton transfer from the water molecule to the second layer of the interstitial oxygen ion")

l. 220: "but easy in reality"

l. 291-292: " to further investigation of the dissolution process to a larger extent"

Jérôme Claverie

Reviewer #3 (Remarks to the Author):

The authors present ab initio molecular dynamics (AIMD) simulations of the dissolution of calcium ions from the (111) surface of calcium silicate. The authors used metadynamics to calculate the free energy change of dissolution of two different calcium ions from the surface. The computational work carried out is impressive. AIMD are very computationally demanding techniques and the authors have combined this with the use of metadynamics. There are a few studies coming out in the literature using this technique, so the work described in this paper is significant. Moreover, the length scale described is quite substantial. I believe that this work is noteworthy as it can provide an atomistic description of the dissolution of this material. The methodology is sound and the results support the conclusions made by the authors.

I have some concerns about the applicability of the research for the wider community especially the experimentalists. The systems described in this paper are very simple and from the pictures provided, seem to be quite small. What is the size of the system being simulated? How can the authors be sure that the free energy change is not an artefact of the small surface area? Would the results be the same if you simulated a system with double the surface area? My concern is that the results could be impacted by the periodic images.

I also have questions surrounding the speciation of the surface. The authors mention that previous studies have shown that there are hydroxyl groups on the surface. How did the authors determine the chemical environment of the surface and therefore justify that the modelled surface is consistent with experimental surface speciation? Moreover, I assume that the pH of the system is 7, therefore, what is the chemical environment of this surface at that pH. Are experiments of dissolution of this material conducted at different pHs.

The authors would like to thank the reviewers for their constructive comments and suggestions. Our detailed responses to the specific points raised are given below. The modifications are in red.

Reviewer #1

Comment 1: The paper “NCOMMS-21-44726, Ab initio mechanism revealing for tricalcium silicate dissolution” starts with an interesting premise: compute the dissolution energy barriers of C_3S using atomistic simulation methods. However, soon in the introduction I have the feeling that the authors are over-selling their research. For instance, starting in line 58: “All these hypotheses and the fit of the calorimetric curve of Ca_3SiO_5 hydration using thermodynamic calculations may be derived from the ignorance on the interfacial reactions during the Ca_3SiO_5 dissolution, especially the dissolution behavior of calcium ions at atomic level. Fortunately, atomistic simulations can tackle these problems.” Atomistic simulation results, including those presented in this paper are truly far from explaining the calorimetric curve, way further than the “thermodynamic calculations may be derived from the ignorance” that the authors mention. As a matter of fact, we don’t even know if dissolution is controlling the hydration of C_3S , or the nucleation of C-S-H is the main mechanism. In addition, they mention “especially the dissolution behavior of calcium ions at atomic level”. Why Ca ions? Why not SiO_4 ions? there is not reason behind this statement.

Response: Thanks for your comments. We fully agree with your statement that the atomistic simulation results cannot comprehensively explain the calorimetric curve of Ca_3SiO_5 hydration, and we are sorry for making you confused by our expression. In fact, we just want to introduce the significance of understanding the interfacial reactions during the Ca_3SiO_5 dissolution. So, we revised the sentence starting in line 58 as follows:

Understanding the interfacial reactions at the water/ Ca_3SiO_5 interface using atomistic

simulations can provide some supplementary and new insights on the Ca_3SiO_5 dissolution.

(lines 59-60, page 3)

In addition, As suggested by the reviewer, we deleted the misleading sentence “especially the dissolution behavior of calcium ions at atomic level”. As for whether the Ca ion or silicate group dissolves first, it has no solid evidence from experimental observation yet. However, the previous MD simulations indicated that the Ca ion desorbs quickly (Manzano et al. ACS Appl. Mater. Interfaces 2015, 7, 27, 14726–14733, Qi et al. Cement and Concrete Research 149 (2021): 106576.) while the silicate group does not dissolve into the bulk water even after a long simulation period and raising the temperature to 700 K (Manzano et al. ACS Appl. Mater. Interfaces 2015, 7, 27, 14726–14733, Sun et al. Cement and Concrete Research, 152 (2022) 106682.). Instead, the silicate groups only stay in their initial positions, rotating their center of mass and contributing the formation of a disordered interphase. In addition, the ζ potential measurements of tricalcium silicate suspensions showed a high concentration of metal ions at the surface, indicating that the dissolution of Ca ion is easier than silicate group too. Theoretically, it is well known that the silicate species have a hydrophobicity nature, especially for silanol groups and siloxane bridges where the water molecules only form one hydrogen bond (Bolis et al. J. Chem. Soc., Faraday Trans., 1991,87, 497-505). On the contrary, the Ca ion in solution is generally six-coordinated with the O ion in water molecules (Manzano et al. Langmuir 2012, 28, 9, 4187–4197). The Ca ion on surface is usually coordinated with less than six O ions, thus the water molecules tend to adsorb on the Ca site (Qi et al. Applied Surface Science, 518 (2020) 146255). In addition, the dissolution process is

usually described by the ligand-exchange model (Werner et al. Colloids Surf., A 1997, 120, 143–146) and there is few ligands coordinated with the silicate group. So, it is unlikely that the Ca_3SiO_5 dissolution begins with the dissolution of silicate group. For a more rigorous proof, we will conduct experiments to get solid observation in the future. But undeniable, there are also silicate groups in Ca_3SiO_5 hydration solution that maybe attributed to the dissolution of Ca making the silicate group isolated with only water molecules surrounding it and more dynamic compared to the initial state. This process is similar to the experimental observation that there is a hydrated layer containing monomeric and subsequently dimeric silicate units around the C_3S surface (Juilland et al. Cement and Concrete Research, 40 (2010) 831-844).

Comment 2: All the text starting in line 65: “Claverie et al.²⁷ investigated the proton transfer at the water/ Ca_3SiO_5 interface using ab initio molecular dynamics (AIMD) simulations and found that the hydroxides formed on the surface are highly stable. However, they did not observe an obvious vertical displacement of Ca ions relative to the initial position. In fact, it is very hard to probe a complete calcium dissolution process at the atomic level even using the traditional molecular dynamics (MD) simulations^{28, 29} with large timescale (i.e. nanoseconds). Moreover, the classical MD is not appropriate to simulate the chemical reaction involving the breakage and formation of bonds. Even using the ReaxFF force field is still not preciously enough to present the reaction pathways for a chemical reaction.” is written to justify the present work and has no fundament. Cleaverie et al. did not observe Ca dissolution, true, and traditional MD cannot reproduce water dissociation, but then, Why “the ReaxFF force field is still not preciously enough to present the reaction pathways for a chemical reaction”? There is no reasoning or citation behind this statement. In fact, ReaxFF

simulations have been the most useful simulation studies so far. They are able to reproduce the dissolution of Ca atoms (see - Manzano, H., Durgun, E., LópezArbeloa, I., & Grossman, J. C. (2015). Insight on tricalcium silicate hydration and dissolution mechanism from molecular simulations. *ACS applied materials & interfaces*, 7(27), 14726-14733, Qi, C., Manzano, H., Spagnoli, D., Chen, Q., & Fourie, A. (2021). Initial hydration process of calcium silicates in Portland cement: A comprehensive comparison from molecular dynamics simulations. *Cement and Concrete Research*, 149, 106576. and Salah Uddin, K. M., & Middendorf, B. (2019). Reactivity of Different Crystalline Surfaces of C₃S During Early Hydration by the Atomistic Approach. *Materials*, 12(9), 1514.) and the accuracy has not been any discussion about the precision of the method. Why is ReaxFF “not precisely enough”? The authors present previous work in the field, but they do not mention several very relevant papers, specially: - Manzano, H., Durgun, E., López-Arbeloa, I., & Grossman, J. C. (2015). Insight on tricalcium silicate hydration and dissolution mechanism from molecular simulations. *ACS applied materials & interfaces*, 7(27), 14726-14733. - Salah Uddin, K. M., & Middendorf, B. (2019). Reactivity of Different Crystalline Surfaces of C₃S During Early Hydration by the Atomistic Approach. *Materials*, 12(9), 1514. The second one is specially relevant, as they use metadynamics to compute dissolution energy barriers. They use ReaxFF instead DFT, but then the methodology is more or less the same as in the present paper. The results are very different, and even if I believe that the present work is more rigorous, the results should be compared and discussed. The simulations of the free energy are interesting, nicely done and nicely presented. As I said, I think that this work is better done than Uddin et al. But I think that there is a mayor methodological problem: why the 111 surface? It is hard to say, but I would say given the structure and symmetry of the MIII polymorph that the created slab is not symmetric. Therefore, how is the plane chosen? Did the authors do any surface reconstruction? are Ca atoms symmetrically distributed in the top and bottom of the slab? The

results are compromised by these factors.

Response: Thanks for your pertinent comments. We deleted the statement on “the ReaxFF force field is still not precisely enough to present the reaction pathways for a chemical reaction” and updated the literature as suggested by the reviewer. We rewrote the paragraph as follows:

Understanding the interfacial reactions at the water/ Ca_3SiO_5 interface using atomistic simulations can provide some new insights on the Ca_3SiO_5 dissolution. The adsorption of water on the Ca_3SiO_5 surface with molecular and dissociative mode²⁵ is the first step of Ca_3SiO_5 hydration, which happens even before contacting the bulk water due to the strongly hydrophilic nature of Ca_3SiO_5 ²⁶. After the surface hydroxylation and the proton hopping into the surface²⁷, the Ca ion will dissolve into the solution destroying the initial surface topology and promoting the further water penetration²⁷, which is a key step for advancing the Ca_3SiO_5 hydration. For this process, the density functional theory (DFT) -based geometry optimization calculations²⁸ indicated the adsorption of water on the Ca ion impairs the bonds strength between the calcium and oxygen ions on the surface. Recently, reactive MD simulations have been widely used to study the Ca_3SiO_5 dissolution and successfully obtain several new perception on dissolution process. Manzano et al.²⁷ found the Ca ion desorbs quickly and tends to accumulate as inner- and outer-sphere complexes at the Ca_3SiO_5 (111) surface. Qi et al.²⁹ showed a more easier Ca dissolution from the Ca_3SiO_5 (010) surface than the Ca_2SiO_4 (100) surface due to the higher surface hydroxylation degree. Sun et al.³⁰ did not observe dissolution of Ca ions from the (010) surface even after 10 ns at 300 K, but after raising the temperature to ~1000 K, the dissolution rate increases five times than that of room temperature. Claverie et al.³¹ first investigated the Ca_3SiO_5 hydration using ab initio molecular dynamics (AIMD) simulations and found that the hydroxides formed on the surface are highly stable. However, they did not observe an obvious vertical displacement of

Ca ions relative to their initial positions. In fact, it is very hard to probe a complete calcium dissolution process at the atomic level using the AIMD simulations^{29,32} with small timescale (i.e. within 100 ps). The breakages of Ca-O_s (O_s indicates the oxygen ion in Ca₃SiO₅) bonds and formation of Ca-O_w (O_w indicates the oxygen ion in water) bonds are indeed a rare event for not only AIMD, but also the reactive MD, which calls for the cooperation with the enhancing sampling method, such as metadynamics. Uddin et al.³³ used reactive forcefield (ReaxFF) combined with metadynamics to calculate the free energy changes of dissolution of Ca ions from various Ca₃SiO₅ surfaces along the reaction coordinate of the distance between the center of mass and the selected calcium atom. However, this collective variable cannot clearly illustrate the nature of dissolution, which is well accepted as a ligand exchange reaction³⁴. Moreover, the chemical reactions at water/Ca₃SiO₅ interface are typically accompanied by electron transfer. Hence, it is indispensable to give an ab initio description of such a fundamental reaction. **(lines 59-91, page 3-4)**

In addition, as we use the different collective variables (or to say the reaction coordinates) and different surfaces with Uddin et al., it is very reasonable to have different results. So, it is meaningless to compare the absolute value between these two works. But as suggested by the reviewer, we still discuss their results in discussion section as follows:

Besides, when the Ca ion is not fully dissolved, the detachment of the Ca ion is spontaneous, while this is reverse as the Ca ion is no more coordinated with O_s. However, as Uddin et al.³³ pointed out that the reactivity of different surfaces for Ca dissolution would be different, thus the value of free energy changes and barriers would vary with surface indexes. **(lines 236-240, page 16)**

As for the choice of the surface, as the reviewer said, it is a very essential methodological

question needed to be solved. The construction of the slab model for MIII polymorph of Ca_3SiO_5 is extremely complicated due to its low symmetry. Therefore, before this work, we had done a lot of work to study the Ca_3SiO_5 surfaces and the corresponding results have been illustrated in detail in our recently published paper and its supplementary information (Li et al. Cement and Concrete Research, 152 (2022) 106684). For more detail, we have cleaved seven low-index surfaces with all possible terminations from the optimized bulk unit cell and calculated the surface energies for these slabs. The positions of the terminations on seven low-index surfaces are shown as follows:

Figure 1. The positions of the terminations on (100) surface. The distances of each termination from the origin are presented at the bottom of each picture. Green sphere - Ca; blue sphere - Si in the center of SiO_4 tetrahedron red sphere - O; yellow plane - the position of the termination.

Figure 2. The positions of the terminations on (010) surface.

Figure 3. The positions of the terminations on (001) surface.

Figure 4. The positions of the terminations on (110) surface.

Figure 5. The positions of the terminations on (101) surface.

Figure 6. The positions of the terminations on (011) surface.

Figure 7. The positions of the terminations on (111) surface.

Then we do the DFT-based surface relaxation. The details for the methodology of this process

(here we choose (111) surface as an example) together with the previous geometry optimization of the Ca_3SiO_5 bulk crystal were added to the Supplementary Method as follows:

DFT-based Geometry optimization method. The geometry optimization of the Ca_3SiO_5 bulk crystal and (111) surface slab were calculated within density-functional theory (DFT), implemented in the Vienna Ab Initio Simulation Package (VASP) code^{1, 2}. The exchange-correlation potential was approximated within the generalized gradient approximation (GGA) using the Perdew-Burke-Ernzerhof (PBE) functional³ due to the outstanding agreement between experimental and theoretical lattice constants⁴ and the good suitability for calculation of calcium silicate species^{5, 6}. The valence electrons of $3s^23p^64s^2$, $3s^23p^2$, $2s^22p^4$, and $1s^1$ were considered for Ca, Si, O and H, respectively. Iterative solutions of the Kohn–Sham equations were expanded in a plane-wave basis set defined by a kinetic energy cutoff of 600 eV⁷. The global convergence criterion of energy for the electronic self-consistent loops was set as 10^{-5} eV. The bulk unit cell was optimized with the tolerance of 10^{-3} eV \AA^{-1} for the ionic relaxation^{6, 8}. Monkhorst-Pack⁹ scheme was used for the k-point sampling with a grid of $3 \times 5 \times 4$ in the first Brillouin zone. For surface relaxation, All surfaces were maintained neutral with integer numbers of basis to preclude the polarizing electric field^{10, 11}. The vacuum was set as 20 \AA with a dipole correction along the z direction to get rid of bogus contributions arising from asymmetry. The two uppermost layers of atoms were completely relaxed while the rest were fixed¹². The lattice constants of slab models were fixed¹³. No symmetry was forced on both sides of slabs. The optimization threshold was 0.02 eV \AA^{-1} for ionic relaxations and the k-points mesh is $2 \times 2 \times 1$.

Then we calculated the surface energy using the following expression:

$$\gamma = \frac{1}{2A}(E_s^{unopt} - NE_b) + \frac{1}{A}(E_s^{opt} - E_s^{unopt}) \quad (1)$$

Where E_s^{unopt} and E_s^{opt} are energies of unrelaxed and relaxed surfaces, N is the number of formula units embodied in the slab, E_b is the bulk energy per formula unit and A is the in-plane area of slab model.

There is a great discrepancy of surface energies between different terminations even in a same miller index (Figure 8). It is known that the temperature needed for calcinating the pure Ca_3SiO_5 is more than 1500 °C (1773 K). According to the definition of the Boltzmann constant (k_B), if we describe the relationship between the motion and energy at the molecular level, 1500 °C can provide 2.48 J/m² ($\gamma_T = k_B T$) for the system. This thermal energy is not only to make the surface cleaved from the bulk crystals, but also for the decomposition of raw materials and phase transitions of Ca_3SiO_5 . From this perspective, we can approximately say the low-index surfaces with various terminations can be easily formed in the C_3S calcination process, but it is difficult to exactly identify which surfaces and terminations will form in high-temperature environment and multi-steps reactions. Here, we choose the (111) surface, which has a greater possibility to form and confirmed by the previous DFT calculations (Durgun et al. J. Phys. Chem. C 2014, 118, 28, 15214–15219) as an example to study the dissolution mechanism. Although the free energy barriers and changes would be various at different surfaces, the nature of dissolution of Ca from C_3S would be similar. In the future, we will continue to get a more comprehensive understanding of the dissolution process on other surfaces and Ca sites, but now this is out of our scope.

Figure 8. Surface energies of seven low-index surfaces with different terminations. The red line represented 2.48 J/m^2 indicates the energy at the C_3S calcination temperature.

We revised the content about choosing the (111) surface as follows:

Because the Ca_3SiO_5 (111) surface has been studied to have a greater possibility to form in practice^{25, 37}, we took the pure Ca_3SiO_5 (111) surface from our previous work²⁵, which was cleaved from the M3 polymorph of Ca_3SiO_5 (obtained from CCSD³⁸), as the M3 polymorph is the most frequently observed in industrial clinkers together with the M1 form³⁹. The details of the DFT-based geometry optimization of the bulk crystal and surface slab were listed in **Supplementary Methods. (lines 276-281, page 18)**

Comment 3: There are several surprising statements when the results are tried to be compared with experiments. For instance: “The mean square displacement (MSD) (Figure 3c) presents a more dynamic property and stronger diffusion ability for the dissolved Ca compared to the previous surface state” Well, of course the atoms will displace more in solution than in the solid, the result has no relevance. As the IR spectra: if there are Si-OH and Ca-OH in the system, the IR spectra will match the experiment, no need to do it. But more surprisingly is the comparison in figure 4f: “From the transmission electron microscopy

(TEM) photograph (figure 4f), we can clearly see when Ca_3SiO_5 encounters with water, the Ca cations are released from the Ca_3SiO_5 surface and form the $\text{Ca}(\text{OH})_2$ in the aqueous solution.” In the TEM image that cannot be seen. Atoms, or complexes, cannot be seen. Those “particles” are way larger than $\text{Ca}(\text{OH})_2$, and they might be more likely C-S-H particles precipitated at the surface or less dense C_3S region due to partial dissolution. I was extremely surprised when I arrived to line 308 and I realise that TEM is from the authors, not from another work. I don’t understand what the TEM is contributing to the paper-

Response: Thanks for your comments. As suggested by the reviewer, **we deleted the content about the MSD and the TEM.** As for the calculated IR spectra, it has several distinctive advantages compared to the experimental IR spectra. One of the most powerful feature for the calculated IR spectra is that we can compute the IR spectra for an isolated component of a mixture system, which would be very useful for the surface system. In the experimental IR spectra, we cannot extract the spectrum of the adsorbent only and omit the contributions from the surface. In addition, our readers are not only the computational chemists, but also the experimentalists. For computational chemists, they are used to get the bond information through visualizing the trajectory. However, for the experimentalists, they would like to obtain some spectroscopy data that can be mutually confirmed with their experimental results. Our IR spectra for the isolated Si-OH and Ca-OH cannot be obtained by the experimental data, thus we think our data would be a useful supplementary information for the experimental IR spectra in Ca_3SiO_5 hydration. If the reviewer still thinks that it is unnecessary to keep this data, we will delete the relevant content later.

Reviewer #2 (Remarks to the Author):

The manuscript NCOMMS-21-44726 « Ab initio mechanism revealing for tricalcium silicate dissolution » presents atomic scale simulations of calcium cations dissolution at the (111)

surface of M3 Ca_3SiO_5 (C_3S in cement chemistry notation). The authors ran well tempered ab initio metadynamics simulations to determine reaction pathways, free energy changes and free energy barriers. The results of this study are originals and of importance for the fundamental understanding of Ca_3SiO_5 dissolution.

In spite of the originality and the considerable work needed for simulations and analysis, several points should be considered to improve the manuscript:

Comment 1: It is not indicated to which oxygen atoms O_s refers to. Generally, O_s refers to oxygen in silicates, but I supposed that here O_s refers to superficial oxygen atoms without discrimination between isolated oxygen anions (O_i) and oxygen in silicates. In both cases, the naming code for chemical species (O_w , O_s) should be provided, and more information should be given on Ca- O_i bonds for which the breakage occurs earlier (l. 106-108 for ex.).

Response: Thanks for your comments. We added the explanation of the O_s , O_w and O_i when they are first shown as follows:

The breakages of Ca- O_s (O_s indicates the oxygen ion in Ca_3SiO_5) bonds and formation of Ca- O_w (O_w indicates the oxygen ion in water) bonds are indeed a rare event for not only AIMD, but also the reactive MD, which calls for the cooperation with the enhancing sampling method, such as metadynamics. (lines 81-84, page 4)

The breakage of the Ca- O_s bond is earlier, but more difficult than with the interstitial oxygen ion (O_i) (from the state C to D to E). (lines 131-132, page 6)

Comment 2: It is also not clear why the (111) surface was chosen for the study. To facilitate the reader's understanding, a top view of this surface before and after protonation occurs, indicating the position of Ca_α and Ca_β , should be provided. It would be also pertinent to

discuss on the difference in the environment of Ca at this surface compared with other ones.

Response: Thanks for your comments. For the choice of (111) surface, we have answered this question in the Response to the Comment 2 of the reviewer 1, For brevity, please kindly see the corresponding answer there.

As suggested by the reviewer, we provided the top view of (111) surface before and after protonation of dangling bonds and indicated the position of Ca_α and Ca_β . Moreover, the coordination environments of Ca at seven low-indexes surfaces were provided in the Supplementary Table 1 and were discussed in a new section “Determination of reaction coordinates and classification of Ca species.” as follow:

Determination of reaction coordinates and classification of Ca species.

The chemical reaction in initial Ca_3SiO_5 hydration, especially the dissolution of Ca ions, is a process of breaking the old Ca-O_s bonds and forming new Ca-O_w bonds. Therefore, we probe into the coordination environment of the Ca ion to calculate the full dissolution pathways. The Ca_3SiO_5 dissolution rates at different surface sites (i.e. flat, step and kink site) are typically different due to the different chemical environments around the Ca ion. The coordination environments of the Ca ions on surfaces are various due to the low symmetry of the M3 type of Ca_3SiO_5 and the large number of possible surfaces formed during the high-temperature calcination process. For example, the Ca coordination environments at seven low-indexes surfaces range from three to seven (Supplementary Table 1). There are four Ca sites in different chemical environment on the Ca_3SiO_5 (111) surface and they can be classified into three- and five- coordinated Ca species, which are indicated by Ca_α and Ca_β in this work, respectively (Figure 1). Because the coordination environments of Ca ions may change the dissolution pathways as well as the thermodynamic and kinetic properties, we investigate the dissolution mechanism for both Ca_α and Ca_β . (lines 102-117, page 5-6)

Figure 1. The top view of initial (111) surface model before (a) and after (b) surface protonation. The green, cyan, yellow, red and white spheres are indicated to the three-coordinated Ca_α species, the five-coordinated Ca_β species, the silicon, oxygen and hydrogen ions, respectively.

Supplementary Table 1. The coordination environments of the Ca ion on other Ca_3SiO_5 surfaces.

Miller index	Ca coordination environment
100	three-, four- and six- coordinated
010	four-, five- and six- coordinated
001	five-coordinated
110	four-coordinated
101	four-, five-, six- and seven- coordinated
011	four- and five- coordinated
111	three- and five- coordinated

Comment 3: Furthermore, the statement: “the results obtained in this work are general and applicable to other kinds of calcium silicate species” (l. 209-210) should be revised. In my opinion, the results are very specific to C_3S and arguably to the surface under study. If not so,

the applicability of the results to other minerals must be proven. The method however can be applied to other minerals.

Response: Thanks for your comments. We have deleted the misleading sentence “the results obtained in this work are general and applicable to other kinds of calcium silicate species”

Comment 4: The formation of stable hydroxyl on the surface before starting the production runs should be discussed because this may influence the dissociation of Ca^{2+} . Moreover, it is not stated when the proton transfer to the O_i of the second layer occurs, and when the water molecule comes to occupy the calcium site. A timeline with the important events could improve the quality of the manuscript.

Response: Thanks for your comments. As said by the reviewer, the situation that the initial configuration of the system influences the outcome of the simulation does exist in the traditional AIMD or MD, due to their insufficient sampling ability. However, metadynamics is an efficient enhancing sampling method for accelerating the MD simulations. Through intermittently introducing an external history-dependent bias potential (in this work are Gaussian kernels) on a few selected degrees of freedom, also called collective variables (CVs), the system is forced to get out of the low energy basin, cross the large energy barrier, and go into more regions on the free energy landscape. Therefore, nearly all the configurations in the designated CVs space can be visited for many times, which are inaccessible in equilibrium MD. In fact, there is no absolute “stable hydroxyl on the surface” in the metadynamics simulations. Through adding a bias potential, the “stable hydroxyl” formed on the Ca^{2+} will dissociate and associate for many times, which guarantees an abundant sampling in the configurational space. For more details, please see the relevant papers about metadynamics (Barducci et al. Phys. Rev. Lett. 100, 020603). Thus, the timeline in the traditional MD/AIMD simulations has no meaning in the metadynamics simulations.

Therefore, we cannot get the information about when the proton first transfer to the O_i of the second layer, and when the water molecule comes to occupy the Ca site. But we can obtain the order of these reactions, as already written in our manuscript.

Comment 5: According to l. 176-177, the regions IV, V and VI are similar to the “perfect” Ca_3SiO_5 . Are they similar in width, in intensity and/or position of the peaks? What about the other regions (I, II, III) ?

Response: Thanks for your comments. We answered all these questions and revised the manuscript as follow :

In the region I (6 - 8.7 Å), it is obvious that the H ion penetrates into the second layer of the surface ($z = 6$ Å), due to the escape of the Ca ion and the proton exchange from water molecules to the inner O_i . Additionally, the position of the peak of O_w and H is overlapped at 8 Å, indicating one water molecule settles down the original position of the dissolved Ca ion with a parallel configuration. These two phenomena are the first time to be observed in the hydration of Ca_3SiO_5 by AIMD simulations. The region II (8.7 - 11.1 Å) is the chemisorbed water molecule region showing an apparent and strong peak for both O_w and H at nearly the same position, which means the chemisorbed water molecules have a well-ordered structure and tend to parallel to the surface. This result is different from our previous work on the perfect Ca_3SiO_5 surface²⁵, which presents a tendency for upright configuration with O_w below H. This divergence indicates the dissolved Ca ion tunes the direction of the chemisorbed water to a flatter configuration. The region III (11.1 - 14.3 Å) is a mixture of the physisorbed water molecule for the surface and the first hydration shell of the dissolved Ca, which expands the destruction area of the layered water structure compared to the surface before dissolution. The region IV (14.3 – 20.8 Å) is the transition layer with a less extent of

structuration oscillating at 0. The region V (20.8 - 28 Å) is the bulk water layer with a density of almost 1, and region VI (28 - 33 Å) is water/vacuum layer. All the above three regions are similar to those on the perfect Ca₃SiO₅ surface not only in width but also in intensity. (lines 192-210, page 13)

Comment 6: More details should be provided with regard to the following sentence on the spontaneous and unspontaneous nature of Ca dissociation: “The thermodynamic and kinetic analyses show that the detachment of Ca is spontaneous when the Ca is not fully dissolved and unspontaneous when Ca is no more coordinated with Os.” (l. 231-233).

Response: Thanks for your comments. When the free energy changes between the two states are negative, we think this process is spontaneous and vice versa. We have described the free energy changes in detail in the results section. In addition, As suggested by the reviewer, we explain this conclusion in the discussion section as follows:

Besides, when the Ca ion is not fully dissolved, the detachment of the Ca ion is spontaneous, while this is reverse as the Ca ion is no more coordinated with O_s. (lines 236-240, page 16)

Comment 7: Other small changes may improve the quality of the manuscript:

- The duration of the production and equilibrium runs should be given in a list or a table to improve the readability (l. 272-275)
- “the M3 type of Ca₃SiO₅ (obtained from CCSD33), which is most frequently observed in industrial clinkers” → “the M3 polymorph of Ca₃SiO₅ (obtained from CCSD33), which is the most frequently observed in industrial clinkers together with the M1 form”
- The region of the zoom in Fig 4f should be reduced accordingly to the scale bar. The snapshot of the simulation is ~1 nm in width.
- Metadynamics simulations applied to calcium dissociation from Ca₃SiO₅ surfaces was

already performed with ReaxFF. A reference to doi:10.3390/ma12091514 would be pertinent.

Response: Thanks for your comments. The duration of the production and equilibrium runs was given in a table as follows:

Table 1. The equilibration and production time for all the simulation projects.

Simulation project	Equilibration time	Production time
Dissolution of Ca_{α} ,		100 ps
Dissolution of Ca_{β} with CN($Ca-O_s$) from 5 to 2		63 ps
Further dissolution of Ca_{β} with CN ($Ca-O_s$) from 2 to 0	At least 2.5 ps	46ps
Equilibrium of final state of Ca_{β}		30ps

The sentence “M3 type of Ca_3SiO_5 (obtained from CCSD), which is most frequently observed in industrial clinkers” was revised according to the reviewer.

As suggested by the reviewer 1, the Fig 4f was removed already.

As also suggested by the reviewer 1, we discussed this paper (doi:10.3390/ma12091514) in our manuscript at this time, for brevity, please kindly find the answers in the Response to comment 1 and 2 for the reviewer 1.

Comment 8: The English language of the manuscript is generally understandable but needs an in-depth revision to improve the quality of the manuscript. E.g. “lead to that” (l. 16); “preciously enough” (l. 72); “any one of” (l. 97)→ “each”; “indicating two water...” (l. 100)→ “indicating that two water”; “rapider” (l.130) → “quicker” or “faster”; “possible”

(l.139; 150) → “likely”; “but also” (l. 160) → “but also because”?; “a further 30 ps equilibrium AIMD simulations” (l.162) → “an additional 30 ps equilibrium AIMD simulation”; “The AIMD also shows that the coordinated Ow with Ca β ” (l.162-163) → “This final run also shows that CN(Ca-Ow)”; “evolution of time” (l. 164) → time evolution; “in the hydration of Ca₃SiO₅ to be observed by the AIMD simulation” (l. 172)→ “of the hydration of Ca₃SiO₅ to be observed by AIMD simulation”; “The region II is the chemisorbed water molecule and III is ... water molecule for the surface” (l. 172-173) → “Water in the region II is chemisorbed and the region III corresponds to ... water molecule on the surface”; “new formed” (l.196) → “newly formed”; “Ca-Os bonds become less” (l.204) → “number of Ca-Os bonds decrease”; “hydron” → “hydrogen” (l. 225); “precious” (l. 238)→ “previous”; l. 260 “the CP2K/Quickstep” → “the CP2K/Quickstep code”; “the Gaussian hills was” (l.289) → “the Gaussian hills were”; “though” → “through” (l.304), “ ω is frequency” (l. 306) → “ ω is the frequency”

Some sentences are hard to understand and should be rephrased/corrected:

- l. 103-106: (e.g. “increase in free energy” should be more appropriate than “increase in free energy changes”) l. 106-108: (e.g “than that with” → “than with”) l. 123-124: “Albeit” (l. 123) → “Although” l. 128-129: Be more specific about the “second free energy minimum”. “pass through state B first”. Why “first”? l. 150: “to exist in reality”, what is meant by “in reality”? l. 152-153 “welcoming one more water molecule” l. 158: “processes the greatest possibility” l. 178-179: “the first and second hydration shells ... are more than” → “the intensity of the peaks which corresponds to the first and second hydration shells...is greater than” l. 181-183: “a more dynamic property and stronger diffusion ability” l. 190-196: the sentence is long and hard to understand in some parts (e.g. “the proton transfer from the water molecule to the second layer of the interstitial oxygen ion”) l. 220: “but easy in reality” l.

291-292: “ to further investigation of the dissolution process to a larger extent”

Response: Thanks for your kind and careful corrections on our manuscript. We revised all the errors and inappropriate expressions as suggested by the reviewer. However, the suggestion “ “increase in free energy” should be more appropriate than “increase in free energy changes” ” would change what we meant to express originally. The free energy change is ΔA , which is different from the free energy A . Thus, we think the original expression is more appropriate.

Reviewer #3 (Remarks to the Author):

Comment 1: The authors present ab initio molecular dynamics (AIMD) simulations of the dissolution of calcium ions from the (111) surface of calcium silicate. The authors used metadynamics to calculate the free energy change of dissolution of two different calcium ions from the surface. The computational work carried out is impressive. AIMD are very computationally demanding techniques and the authors have combined this with the use of metadynamics. There are a few studies coming out in the literature using this technique, so the work described in this paper is significant. Moreover, the length scale described is quite substantial. I believe that this work is noteworthy as it can provide an atomistic description of the dissolution of this material. The methodology is sound, and the results support the conclusions made by the authors. I have some concerns about the applicability of the research for the wider community especially the experimentalists. The systems described in this paper are very simple and from the pictures provided, seem to be quite small. What is the size of the system being simulated? How can the authors be sure that the free energy change is not an artefact of the small surface area? Would the results be the same if you simulated a system with double the surface area? My concern is that the results could be impacted by the periodic images.

Response: Thanks for your comments. The size of the systems for metadynamics simulations is $14.21 \text{ \AA} \times 11.72 \text{ \AA} \times 36 \text{ \AA}$ and for equilibrium AIMD is $14.21 \text{ \AA} \times 11.72 \text{ \AA} \times 48 \text{ \AA}$. These systems are not small for the AIMD simulations with such a long simulation time (totally more than 200 ps). Here are some outstanding papers regarding the AIMD simulations, in which the size of the systems can be a reference for the reviewer (Liu et al. ACS Catal. 2021, 11, 19, 12336–12343; Chen et al. Nat Commun 12, 3725 (2021); Li et al. Nat. Mater. 18, 697–701 (2019)). For precluding the periodic images effect, we followed the method suggested by Wang et al. (Wang et al. J. Am. Chem. Soc. 2013, 135, 29, 10673–10683.) that the distance between the targeted object and itself in the next periodic image should be equal to or larger than about 5 \AA . Moreover, we did not observe the targeted Ca ion is affected by itself in the next periodic image. As for the surface area, it is well known that the surface coverage of adsorbents will influence the adsorption energy. However, in this work, the surface coverage degree of water molecules is one hundred percent. Thus, under the condition that the periodic images effect is removed, no matter how large the slab, it will not influence this value. As for whether our free energy surface would change as the surface area becomes larger, it really needs to test. In fact, we have followed the reviewer's suggestion and doubled our models to test whether the free energy surface will change. But, when the system becomes double, the time for computation increased more than fourfold. Because it only makes sense when we compare two converged free energy surfaces, we must get a converged free energy surface using this doubled model, which is really computationally demanding. We will continue to run this doubled model and study the surface area effect on free energy changes in detail in the future.

Comment 2: I also have questions surrounding the speciation of the surface. The authors mention that previous studies have shown that there are hydroxyl groups on the surface. How

did the authors determine the chemical environment of the surface and therefore justify that the modelled surface is consistent with experimental surface speciation? Moreover, I assume that the pH of the system is 7, therefore, what is the chemical environment of this surface at that pH. Are experiments of dissolution of this material conducted at different pHs.

Response: Thanks for your comments. Some previous experimental studies determined the surface speciation and we have considered these in our surface models. For example, the infrared (IR) spectroscopy and inelastic neutron spectroscopy (INS) proved that there are Ca-OH bonds on the Ca_3SiO_5 surfaces (Thissen et al. *Chemistry–A European Journal* 24, 8603-8608 (2018).); Thomas et al. *Chemistry of materials* 15, 3813-3817 (2003)). In addition, The $^{29}\text{Si}\{^1\text{H}\}$ cross polarization magic angle spinning nuclear magnetic resonance (CPMAS NMR) and attenuated total reflection-Fourier transform infrared (ATR-FTIR) identified the existence of Si-OH bonds in initial period of Ca_3SiO_5 hydration (Pustovgar et al. *Nature communications* 7, 1-9 (2016); Bellmann et al. *Cement and Concrete Research* 40, 875-884 (2010); Brough et al. *Journal of Materials Science* 29, 3926-3940 (1994); Higl et al. *Cement and Concrete Research* 142, 106367 (2021)). In fact, the surface of Ca_3SiO_5 can be partially hydroxylated before contacting the bulk water due to the strongly hydrophilic nature of Ca_3SiO_5 and humidity in the air. Thus, before adding bulk water on the Ca_3SiO_5 surface models, we first put isolated water molecules to all the possible adsorption sites and did the DFT-based geometry optimization. For more details about this process, please refer to our previous work (Li et al. *Cement and Concrete Research*, 152 (2022) 106684). After that, we only preserved the water molecules with dissociative adsorption on the surface. The dissociated H ions came to Si-O groups forming Si-OH bonds, while the dissociated OH bonds came to the Ca ion on the surface forming Ca-OH bonds. Then we put bulk water on the surface and ran the AIMD simulations. During AIMD, the chemical environment was dynamic, which means the initial state we set will change constantly. Even if we preserved

the OH on the surface initially, it still could associate one H from other water molecules reforming a water molecules and desorbed on the surface. At the same time, there were also other newly formed OH on the surface. Thus, it is not very important to set the chemical environment artificially because the chemical environment will evolve as the AIMD proceeds. What's more, the metadynamics has a strong sampling ability, which can sample nearly all the chemical environment around the targeted Ca ion.

As for the pH value, it can be a microscopic or macroscopic quantity, which is difficult to be considered at atomistic scale. It is unreasonable to compute the pH value only using hundreds of water molecules and several H⁺ or OH⁻. In fact, the influence of pH on the experimental results of dissolution can be interpreted as the influence of the H⁺ or OH⁻ on the targeted Ca ion. In the AIMD, the dissociation of the water and association of the H⁺ and OH⁻ are “on-the-fly” as the evolution of the simulations. For example, there is no OH⁻ around the targeted Ca ion initially, but at the end of the AIMD, the stable state of the system is the dissolved Ca ion coordinated with five water molecules and one hydroxyl group. And it is impossible to calculate the pH value at this state due to very limited atoms of the AIMD systems. In addition, the Ca₃SiO₅ hydration is conducted at pH = 7, and the change of pH value of the Ca₃SiO₅ hydration is the consequence of the surface reaction, not by the human intervention. The dissolution of the Ca²⁺ will make the H⁺ adsorbed on the surface and leave the OH⁻ in the solution, thus the pH value will increase from 7 to ~ 12.5 gradually (Kumar, Synthetic calcium silicate hydrates, EPFL, 2017) and our simulations are consistent with this phenomena. However, the reviewer's suggestion inspires us to continue simulate the dissolution with different H⁺ or OH⁻ around the targeted Ca ion to study whether the free energy or mechanism will change with this variable in the future.

REVIEWER COMMENTS

Reviewer #1 (Remarks to the Author):

I want to thank the authors for such a clear response to my questions. Technical details are clear now, and I don't have any further question in that sense.

On the bad side, I still consider that despite the high quality of the calculations, the information that they obtain and present is not relevant in a real world scenario. From a very fundamental scientific point of view I like the paper, but I do not expect any real impact. Cement hydration cannot be represented by a single Ca atom desorption, and there is no explanation on how they could possibly upscale their result.

In any case, nice work.

Reviewer #2 (Remarks to the Author):

The authors addressed all the issues with care. However, some points still need attention:

1) The meaning of O_i , which corresponds to isolated oxygen anions (called interstitial oxygen ions in the manuscript), was clarified in the manuscript. However, I find still unclear what O_s refers to: " O_s indicates the oxygen ion in Ca_3SiO_5 " (l. 85 p. 4). Is it oxygen in SiO_4^{4-} or oxygen in C3S without discrimination between oxygen atoms in SiO_4^{4-} and isolated O^{2-} ? If it is oxygen in SiO_4^{4-} , why $CN(Ca-O_i)$ was not chosen as a variable?

2) Indeed, a timeline for metadynamics simulation, where a bias potential is applied, has no physical meaning. However, for informative purpose I think it should be interesting to highlight the sequence of the important events and in which run they occur (maybe by adding a column in the Table 1?), remembering that the simulation is biased and accelerated. If there is "no stable hydroxyl groups" on the surface, please explain what are the protons represented in Figure 1 (b).

The authors wrote "The breakage of the Ca-Os bond is earlier, but more difficult than with the interstitial oxygen ion (O_i)" (l. 136 p.7). I think the terms "earlier" and "more difficult" should be corrected and/or better explained.

3) I think that the sentence "Besides, when the Ca ion is not fully dissolved, the detachment of the Ca ion is spontaneous, while this is reverse as the Ca ion is no more coordinated with Os. " (l.242, p.16) is a repetition of "The thermodynamic and kinetic analyses show that the detachment of Ca is spontaneous when the Ca is not fully dissolved and unspontaneous when Ca is no more coordinated with Os.", which appears below and is better worded. So I think the later is not necessary. I still think that this sentence need to be clarified. Saying that the detachment of Ca is unspontaneous when Ca is no more coordinated with Os (CN(Ca-Os)=0) does not make sense to me because once the detachment occurred, there is no reason to say that it is spontaneous or not (maybe the authors are speaking about the reverse reaction?). Maybe this sentence should be better explained in terms of CN(Ca-Os) and CN(Ca-Ow)?

4) The terms "parallel configuration" (l. 201, p.13) and "parallel to the surface" (l 206,p.13 and l.229,p.14) for water molecules could be clearer saying "oriented with their dipole moment perpendicular to the surface". Otherwise, "dissolved Ca ion tunes the direction of the chemisorbed water to a flatter configuration" means that Ca dissolution lead to water molecules oriented with their dipole perpendicular to the surface? The explanation of the regions observed in the atomic density profile was improved. However the authors should explain what is "the destruction area of the layered water structure"? (l. 211, p. 13). In addition, maybe the term "perfect Ca₃SiO₅" should be explained as a surface which has not experienced Ca dissolution?

5) For the sake of clarity, it should be good to explain the notation of the states (e.g. X(CN(Ca-Os), CN(Ca-Ow)) at the beginning of the "Results" section.

6) "Claverie et al.³¹ first investigated the Ca₃SiO₅ hydration using ab initio molecular dynamics (AIMD) simulations and found that the hydroxides formed on the surface are highly stable." has to be changed to "Claverie et al.³¹ first investigated the Ca₃SiO₅ hydration using ab initio molecular dynamics (AIMD) simulations and found that the hydroxides formed on superficial oxide ions are highly stable."

Some parts should be rephrased to improve the English language and/or the understanding of the manuscript:

"These two phenomena are the first time to be observed in the hydration of Ca₃SiO₅ by AIMD simulations." (l. 202 p. 13)"

"a less extent of structuration oscillating at 0" (l. 212 p. 13)

"surface indexes" (l.245, p. 16) → "Miller indices"

"is most likely" (l. 178, p. 11) → "is the most likely"

"is most stable" (l.186, p.11) → "is the most stable"

"Because the coordination environments of Ca ions may change the dissolution pathways as well as the thermodynamic and kinetic properties. Thus..." → "The coordination environments of Ca ions may change the dissolution pathways as well as the thermodynamic and kinetic properties. Thus..." (l. 144, p. 11)

"can be interpreted as a process that" (l. 226, p. 14) → "can be interpreted as a process where" (l. 226, p. 14)

"to further investigation" (l. 333, p. 20) → "for further investigation"

Jérôme Claverie

Reviewer #3 (Remarks to the Author):

The authors has satisfied all my concerns.

The authors would like to thank the reviewers for their constructive comments and suggestions. Our detailed responses to the specific points raised are given below. The modifications are in red.

Reviewer #1

I want to thank the authors for such a clear response to my questions. Technical details are clear now, and I don't have any further question in that sense. On the bad side, I still consider that despite the high quality of the calculations, the information that they obtain and present is not relevant in a real world scenario. From a very fundamental scientific point of view I like the paper, but I do not expect any real impact. Cement hydration cannot be represented by a single Ca atom desorption, and there is no explanation on how they could possibly upscale their result. In any case, nice work.

Reviewer #2

The authors addressed all the issues with care. However, some points still need attention:

Comment 1: The meaning of O_i , which corresponds to isolated oxygen anions (called interstitial oxygen ions in the manuscript), was clarified in the manuscript. However, I find still unclear what O_s refers to: " O_s indicates the oxygen ion in Ca_3SiO_5 " (l. 85 p. 4). Is it oxygen in SiO_4^{4-} or oxygen in C_3S without discrimination between oxygen atoms in SiO_4^{4-} and isolated O^{2-} ? If it is oxygen in SiO_4^{4-} , why $CN(Ca-O_i)$ was not chosen as a variable?

Response: Thanks for your comments. In this work, the O_s refers to the oxygen ions in C_3S without discrimination between oxygen ions in SiO_4^{4-} and isolated O^{2-} . So, we revised the original sentence " O_s indicates the oxygen ion in Ca_3SiO_5 " as follows:

O_s indicates all the oxygen ion in Ca_3SiO_5 (lines 78-79, page 4)

Comment 2: Indeed, a timeline for metadynamics simulation, where a bias potential is applied, has no physical meaning. However, for informative purpose I think it should be interesting to highlight the sequence of the important events and in which run they occur (maybe by adding a column in the Table 1?), remembering that the simulation is biased and accelerated. If there is "no stable hydroxyl groups" on the surface, please explain what are the protons represented in Figure 1 (b).

Response: Thanks for your comments. The sequences of the important events in the three simulations have been highlighted in Figures 2b, 3b and 4a by arrows and in Figures 2d, 3d and 4b by flow charts. In addition, because the Table 1 is in Methods section, which is expected to contain the information of methods instead of the results of simulations, it may be not appropriate to repeat highlighting the sequences of the important events through adding a column in the Table 1. As for the protons represented in Figure 1 (b), they were dissociated from the isolated water molecule through the DFT-based geometry optimization calculations. As written in the Methods section, we firstly adsorbed isolated water molecules to saturate the dangling bond on the Ca_3SiO_5 surface considering that the adsorption of water molecule on the Ca_3SiO_5 surface occurs even before contacting bulk water. During AIMD, the bond formation and breakage were dynamic, which means the initial state we set will change during simulations. Even if we get stable hydroxyl groups on the surface initially through the DFT-based geometry optimization, they will still be possible to dissociate from the surface and reformed on the surface.

Comment 3: The authors wrote "The breakage of the Ca-O_s bond is earlier, but more difficult than with the interstitial oxygen ion (O_i)" (l. 136 p.7). I think the terms "earlier" and "more difficult" should be corrected and/or better explained.

Response: Thanks for your comments. We revised the corresponding sentence as follows:

The breakage of the bond between Ca and O_{si} (the oxygen ion from the silicate group in Ca_3SiO_5) is earlier than that between Ca and O_i (the interstitial oxygen ion in Ca_3SiO_5) due to the sequence of reaction pathways from state C to D to E. While the breakage of Ca- O_{si} bond is more difficult than that of Ca- O_i bond owing to the higher free energy barriers between states C and D. (lines 128-132, page 6)

Comment 4: I think that the sentence "Besides, when the Ca ion is not fully dissolved, the detachment of the Ca ion is spontaneous, while this is reverse as the Ca ion is no more coordinated with O_s . " (1.242, p.16) is a repetition of "The thermodynamic and kinetic analyses show that the detachment of Ca is spontaneous when the Ca is not fully dissolved and unspontaneous when Ca is no more coordinated with O_s .", which appears below and is better worded. So I think the latter is not necessary. I still think that this sentence needs to be clarified. Saying that the detachment of Ca is unspontaneous when Ca is no more coordinated with O_s ($CN(Ca-O_s)=0$) does not make sense to me because once the detachment occurred, there is no reason to say that it is spontaneous or not (maybe the authors are speaking about the reverse reaction?). Maybe this sentence should be better explained in terms of $CN(Ca-O_s)$ and $CN(Ca-O_w)$?

Response: Thanks for your comments. The criterion for determining whether the dissolution of Ca is spontaneous or not using whether Ca fully dissolves from the surface is indeed not appropriate and misleading. Thus, we rewrote the misleading sentences about the spontaneous and unspontaneous nature of Ca dissociation in the Discussion section and conclusion part as follows:

Besides, the free energy changes for the detachment of Ca_α is negative, while for the detachment of Ca_β is positive, which indicates that dissolution of Ca_α is spontaneous and Ca_β is unspontaneous. (lines 237-239, page 16)

The thermodynamic and kinetic analyses show that the detachment of Ca_α is spontaneous, while the detachment of Ca_β is unspontaneous. (lines 264-265, page 17)

Comment 5: The terms "parallel configuration" (l. 201, p.13) and "parallel to the surface" (l. 206, p.13 and l.229, p.14) for water molecules could be clearer saying "oriented with their dipole moment perpendicular to the surface". Otherwise, "dissolved Ca ion tunes the direction of the chemisorbed water to a flatter configuration" means that Ca dissolution lead to water molecules oriented with their dipole perpendicular to the surface? The explanation of the regions observed in the atomic density profile was improved. However the authors should explain what is "the destruction area of the layered water structure"? (l. 211, p. 13). In addition, maybe the term "perfect Ca_3SiO_5 " should be explained as a surface which has not experienced Ca dissolution?

Response: Thanks for your comments. We revised the inappropriate expressions as follows:

Additionally, the position of the peak of O_w and H is overlapped at 8 Å, indicating one water molecule settles down the original position of the dissolved Ca ion with its dipole moment parallel to the surface. (lines 193-195, page 13)

The region II (8.7 - 11.1 Å) is the chemisorbed water molecule region also showing an apparent and strong peak for both O_w and H at nearly the same position, which means the chemisorbed water molecules have a well-ordered structure and tend to orient their dipole moments parallel to the surface. (lines 198-201, page 13)

In this region, the obvious overlap of the peaks for O_w and H disappears, indicating a destruction of the layered water structure. (lines 205-207, page 13)

This result is different from our previous work on the Ca_3SiO_5 surface without Ca dissolution.

(lines 201-202, page 13)

All the above three regions are similar to those on the Ca_3SiO_5 surface without Ca dissolution not only in width but also in intensity. (lines 209-211, page 13)

Comment 6: For the sake of clarity, it should be good to explain the notation of the states (e.g. $X(\text{CN}(\text{Ca-O}_s), \text{CN}(\text{Ca-O}_w))$) at the beginning of the "Results" section.

Response: Thanks for your comments. We explained the notation of the states at the beginning of the "Results" section as follow:

The coordinate of the state is present in form of $X(\text{CN}(\text{Ca-O}_s), \text{CN}(\text{Ca-O}_w))$, where X indicates the state number on the FES. (lines 119-120, page 6)

Comment 7: "Claverie et al.³¹ first investigated the Ca_3SiO_5 hydration using ab initio molecular dynamics (AIMD) simulations and found that the hydroxides formed on the surface are highly stable." has to be changed to "Claverie et al.³¹ first investigated the Ca_3SiO_5 hydration using ab initio molecular dynamics (AIMD) simulations and found that the hydroxides formed on superficial oxide ions are highly stable."

Some parts should be rephrased to improve the English language and/or the understanding of the manuscript:

"These two phenomena are the first time to be observed in the hydration of Ca_3SiO_5 by AIMD simulations." (l. 202 p. 13)"

"a less extent of structuration oscillating at 0" (l. 212 p. 13)

"surface indexes" (l.245, p. 16) → "Miller indices"

"is most likely" (l. 178, p. 11) → "is the most likely"

"is most stable" (l.186, p.11) → "is the most stable"

"Because the coordination environments of Ca ions may change the dissolution pathways as well as the thermodynamic and kinetic properties. Thus..." → "The coordination environments of Ca ions may change the dissolution pathways as well as the thermodynamic and kinetic properties. Thus..." (l. 144, p. 11)

"can be interpreted as a process that" (l. 226, p. 14) → "can be interpreted as a process where" (l. 226, p. 14)

"to further investigation" (l. 333, p. 20) → "for further investigation"

Jérôme Claverie

Response: Thanks for your kind and careful corrections on our manuscript. We revised all the errors and inappropriate expressions as suggested by the reviewer.

Reviewer #3:

The authors has satisfied all my concerns.